# Information-guided Planning: An Online Approach for Partially Observable Problems

**Matheus Aparecido do Carmo Alves**
Lancaster University
Lancaster, United Kingdom
m.a.docarmoalves@lancaster.ac.uk

**Amokh Varma**
Indian Institute of Technology
Delhi, India
amokhvarma@gmail.com

**Yehia Elkhatib**
University of Glasgow
Glasgow, United Kingdom
yehia.elkhatib@glasgow.ac.uk

**Leandro Soriano Marcolino**
Lancaster University
Lancaster, United Kingdom
l.marcolino@lancaster.ac.uk

## Abstract

This paper presents IB-POMCP, a novel algorithm for online planning under partial observability. Our approach enhances the decision-making process by using estimations of the world belief's entropy to guide a tree search process and surpass the limitations of planning in scenarios with sparse reward configurations. By performing what we denominate as an *information-guided planning process*, the algorithm, which incorporates a novel I-UCB function, shows significant improvements in reward and reasoning time compared to state-of-the-art baselines in several benchmark scenarios, along with theoretical convergence guarantees.

## 1 Introduction

Decision-making agents are increasingly being used under uncertainty in nontrivial systems [17]. Based on the available information, such agents must evaluate potential decisions in the current environment to build a robust plan that either accomplishes the task or leads to a better situation. Consider, for instance, a scenario in which drones are deployed to rescue people in a hazardous environment. Drones need to quickly identify critical locations where they can offer support and save lives. If no clear target is found, they must rapidly formulate a plan based on their current knowledge and observable information to find an area in need of assistance from their current position.

The above context describes a partial observability problem that is recurrent in the literature [2, 7, 12]. The Partially Observable Monte-Carlo Planning (POMCP) [32] algorithm is commonly suggested as a method to address these problems because it enables agents to perform planning in an online manner while handling uncertainty [6, 36, 38]. However, several state-of-the-art solutions that rely on POMCP often struggle when rewards are delivered sparsely in time or outside their reasoning horizon. For example, in foraging, there may be no clear target to spot from the current observation and/or estimated map so far. Here, we find a fundamental academic challenge: How can we improve agent performance when rewards are delivered sparsely in time or out of their reasoning horizon?

Every time an agent makes an observation of the world, it gains information that can be used to improve its internal planning process. Based on this, some works improve the traditional planning process by handling the lack of prior information using the approximation of the world's dynamic models [16, 27], embedding supportive planning methods within the tree [2, 7], extracting non-explicit information from the observations [6, 16], enhancing the method's inner calculations [1, 8, 26] or employing neural networks-based techniques to improve the search quality [18, 37].

37th Conference on Neural Information Processing Systems (NeurIPS 2023).

Overall, these strategies can improve search quality within a defined simulation horizon by using extensions of traditional partially observable models and enabling the integration of additional knowledge into planning. However, such knowledge may not be available to agents beforehand (e.g., conditional observation probabilities), or may demand significantly more computational resources to perform the task (e.g., time and memory). While it is important to note that incorporating more knowledge benefits efficiency in terms of objective accomplishment, it does not resolve challenges in sparse-reward scenarios, as most of the solutions remain reward-driven.

Hence, we present *Information-based POMCP (IB-POMCP)*, an online planning algorithm that uses a novel *information framework* to boost the decision-making process under partial observability even when no rewards are available in the reasoning horizon. Our framework refines the traditional UCB1 action selection strategy by implementing our proposed *Information-guided UCB (I-UCB)* function, which is capable of leveraging entropy and estimating information gain, using both real-world and internally generated observations to identify promising states in the search process that leads the agent to quickly accomplish its objective. Its application together with our new particle filter reinvigoration strategy, which considers the current system's entropy to calibrate the reinvigoration, indicates that this approach may lead the agent to act hybridly when solving a partially observable problem by better weighing the exploration, exploitation, and information levels within the tree. We ran experiments across five benchmarks and compared them with state-of-the-art baselines, obtaining significantly higher average rewards (up to 10 times) while performing faster reasoning (up to 93%).

## 2 Related Work

In the current state-of-the-art, POMCP [32] remains a relevant solution due to its adaptability and problem-solving capability across the most diverse domains [3, 23, 24, 41]. However, many proposals are tailored to specific problems, leaving room for improvement, especially when handling sparse reward scenarios. Our approach addresses this gap by shifting POMCP's reward-based planning to our "*information-guided planning strategy*", capable of leveraging observation space entropy (information) and guiding the agent to promising spots from where it solves such scenarios.

Other methods suggest approximating the latent world functions to handle the lack of knowledge [11, 13, 19, 35]. However, these approaches are directly bounded by the belief in space dimensionality, which grows exponentially and hence requires exponentially more resources as the complexity of the problem increases [17]. We propose the extraction of information using only the observations received from the simulations, improving the reasoning without requiring significantly more resources.

Some studies recommend belief-dependent POMDP models to improve POMCP's reasoning capabilities [4, 21]. Under certain assumptions, these models have shown great efficiency when solving problems with large action and observation spaces. A recent advancement is $\rho$-POMCP [36], an efficient online planner that boosts the POMCP's search process by estimating belief-dependent rewards. However, this method relies on explicit access to the observation and transition function, struggling while reasoning under time constraints. TB $\rho$-POMCP [36] tries to handle scenarios with limited time to reason, penalising efficiency in order to do so. IB-POMCP can improve planning using less information and without significant penalties for reasoning time.

Time-constrained problems are prevalent in the evaluation of online planning algorithms. In real-time strategy games, a common strategy is to use macro and micro predefined coordination scripts to support the decision-making [25, 40]. Even though they handle several levels of coordination, these methods still require behavioural templates to plan. Similar problems arise in the multi-agent swarm research [28] where they avoid developing a complex planning process in order to save time (using, for example, Flat Monte Carlo approaches). We propose the execution of efficient planning under partial observability from scratch and without penalising the planning quality.

The Upper Confidence Bound (UCB1) [20] is a relevant algorithm in the literature for evaluating and selecting actions in online planning solutions and is recurrently used inside Monte-Carlo decision-making frameworks [15, 29, 35]. While we acknowledge the existence of other action selection methods available in the literature, such as the Thompson Sampling method [5] and the PUCT algorithm [33], in this paper, we focus on augmenting the POMCP's capabilities by modifying the UCB1. Differently from PUCT (*AlphaGo*) [33], for example, our proposed *I-UCB* well-fits partially observable scenarios while enhancing POMCP planning capabilities through an entropy-based perspective, without relying on pre-trained models or a large amount of data.

Recently, MENTS and ANTS improved the Monte-Carlo Tree Search (MCTS) performance using maximum entropy policy optimisation networks [18, 37]. In contrast to both works, we focus our study on partially observable models and, instead of directly optimising latent functions, we improve planning by embedding knowledge into the search process. Furthermore, both methods rely on prior problem-specific knowledge and data to train neural networks, including information on the maximum episode length, which helps them handle issues like sparse rewards. However, we assume that the agent must reason under a limited number of steps (often lower than necessary to accomplish the problem) and execute the problem from scratch, carrying no knowledge between executions.

Finally, although we focus on improving POMCP, our contributions can also benefit other algorithms, such as DESPOT-based solutions [34]. DESPOT's main focus is on filtering a large observation space to plan using a sparse approximation of the problem, hence, a smaller reasoning space. However, the algorithm still struggles in large action spaces, and/or sparse reward scenarios where the optimal policy could be long (requiring many steps). Our proposal addresses POMDPs featuring sparse rewards in an efficient manner while running the search process from scratch.

## 3   Background

**Markovian Models –** The Markov Decision Process (*MDP*) is a mathematical framework to model stochastic processes in a discrete-time flow. The model specifies that each process is composed of a set of states $\mathbf{S}$, a set of actions $\mathbf{A}$, a reward function $\mathcal{R} : \mathbf{S} \times \mathbf{A} \to \mathbb{R}$, and a transition function $\mathcal{T}$. On the other hand, the Partially Observable MDP (*POMDP*) extends the MDP representation to problems in which the agent cannot directly access the current state. Instead, the agent receives an observation $z \in \mathbf{Z}$, determined by observation probabilities $\mathcal{Z}^a_{s,z} = P(z_{t+1} = z \mid s_{t+1} = s; a_t = a)$. The belief function $\mathcal{B}(h)$ is a distribution over the state space $\mathbf{S}$, which describes the conditional probability of being in a particular state $b \in \mathbf{S}$ given the current history $h$, $\mathcal{B}(h, b) = P(s_t = b \mid h_t)$, where the history $h \in \mathbf{H}$ is a sequence of action-observation pairs $h_t = \{a_0, z_0, \ldots, a_t, z_t\}$. The decision process follows a policy $\pi(h, a) = P(a_{t+1} = a | h_t)$, while choosing actions.

**Monte-Carlo Tree Search (MCTS) –** MCTS algorithms aim to determine the optimal action $a^*$ for a given state $s$ by simulating the world steps within a tree structure. Each node in the Monte-Carlo tree $\mathbf{T}$ is represented by $(s, \mathcal{V}, \mathcal{N})$, i.e., a tuple with a *state* $s$, a *value* $\mathcal{V}(s, a)$, and a *visitation count* $\mathcal{N}(s, a)$ for each action $a \in \mathbf{A}$. The *value* of the node represents the expected cumulative reward for the simulated states. The number of visits to the state $s$ is represented by $\mathcal{N}(s) = \sum_{a \in \mathbf{A}} \mathcal{N}(s, a)$. The simulations are divided into two main stages: *search* and *rollout*. Each state in the search tree is viewed as a multi-armed bandit taking actions usually chosen by the Upper Confidence Bound (UCB1) algorithm. *UCB1 is a well-known* algorithm that tries to increase the value of less-explored actions by attaching a bonus inversely proportional to the number of times each action is tried, following the update-equation $UCB1(s, a) := \mathcal{V}(s, a) + c\sqrt{\frac{log(\mathcal{N}(s))}{\mathcal{N}(s,a)}}$. The scalar $c$ is the exploration constant responsible for weighting the exploration value within the UCB1 function. With a correct definition of $c$, the value function converges in probability to the optimal value, $\mathcal{V}(s, a) \xrightarrow{p} \mathcal{V}^*(s, a)$.

**Partially Observable MC Planning (POMCP) –** POMCP is an extension of MCTS for partially observable problems. Analogously to it, each node is represented by a tuple $(h, \mathcal{V}(h), \mathcal{N}(h), \mathcal{F}_h)$ and the algorithm is divided into the search and rollout stages. In contrast, *POMCP* uses an unweighted particle filter $\mathcal{F}_h$ to approximate the belief state at each node $h$. This strategy requires a *Monte-Carlo simulator* $\mathcal{G}$, which can sample a state $s'$, reward $r$ and observation $z$ given a state-action pair. Recursively down the tree, all sampled information is added to a *particle filter* $\mathcal{F}$. At the end of the search stage, the best action is chosen and a real observation is received from the environment. Considering the current action-observation pair, we move to the new root node and perform the *particle reinvigorating process* [32], which is responsible for updating and boosting the new root's particle filter belief state approximation. After that, we continue the planning procedure (starting the search process) as described above. We refer the reader to Silver and Veness (2010) for further detail.

**Shannon entropy in Information Theory –** A traditional approach to measure uncertainty in stochastic processes is to use Shannon's Entropy function $\mathcal{H}(X) = -\sum_{i=1}^n P(x_i) \log(P(x_i))$, where $X$ is a discrete random variable and $x_i$ are the possible outcomes from $X$. The core idea is to find a reliable measure for "information", that is, to calculate the degree of *surprise* of an event given the available space of possibilities. If an event often occurs, the information inside the event is likely

not "novel". Hence, there is no "surprise" when it occurs. On the other hand, if an event rarely occurs, the "surprise" is greater. Comparing both situations, we can follow events with higher surprise levels to explore the space of possibilities, and safely follow events with lower surprise levels to exploit the environment, hence, we can adapt an agent's behaviour according to the entropy of the current state.

## 4 Information-based Partially Observable Monte Carlo Planning

**Algorithm Outline –** Our algorithm starts with the *(i) initialisation of our tree structure*, where we create our root and calculate the probability of stepping into this node given our current knowledge. Since the initial knowledge about the problem is assumed to be none, we can not calculate this probability, hence, we initialise it as 0. Subsequently, we *(ii) initialise the particle filter of our root node*, generating possible states from which we can simulate the agents' actions and search for a solution to the target problem. Initially, the beliefs are generated through a uniform distribution.

Our search process is iterative, as commonly found in the literature [32, 39, 2], where we sample a state from our current belief to perform multiple rounds of simulation. However, unlike the typical approach, we propose the implementation of an adaptive exploration coefficient and of our novel I-UCB function instead of the usual UCB1. Therefore, at every iteration of our search (before starting a simulation), we first perform our *(iii) exploration coefficient adaptation*, adjusting it using the entropy (level of information) estimated over the set of observations collected through the simulations. Then, with the updated coefficient in hands, we start our *(iv) simulation with the I-UCB function*. While choosing actions, our proposal searches for solutions and takes actions based on observations' entropy besides the collection of rewards. At the end of each simulation, we re-update the exploration coefficient based on the information gained during the path traversal. When we finish the search, we *(v) select the best action* based on the estimated rewards, entropy, and the number of visits of each action's node to determine the most promising path to follow in the real environment.

After taking an action and receiving a new real observation from the environment, we restart the IB-POMCP algorithm. However, the initial steps are slightly changed to maintain the knowledge and update the current information, i.e., perform online planning. Unlike the first iteration, we now *(i) update the root node* by traversing the tree based on the action taken and the most recent observation from the environment. Consequently, we recalculate the probability of stepping into the new root node using the information available in the tree. With the probability in our hands, we now *(ii) update the particle filter of our root node* by reinvigorating our belief, generating new states using the uniform distribution while maintaining promising states for simulation. Then, after performing these updates, we restart the search and repeat the whole process. Now, let us discuss these steps in detail.

**(i) Root Initialisation and Update –** This step is responsible for maintaining the tree structure $\mathbf{T}$ coherent with the world and the information gained while performing online planning. Consider $\mathfrak{r}$ as the tree $\mathbf{T}$'s root node, $h_{\mathfrak{r}}$ as the history of the root $\mathfrak{r}$ (our current history), $a$ as the action taken by the agent in the real world and $z$ as the most recent observation received from the real world (after the agent's action). **As in POMCP**, this step considers three possible procedures:

• **Initialisation:** *if $T$ does not exist,* we create a root node $\mathfrak{r}$ using the current history $h_{\mathfrak{r}}$, else;
• **Update:** *if $T$ exists and the nodes $h_{\mathfrak{r}}a$ and $h_{\mathfrak{r}}az$ also exist,* we walk in the existing tree, traversing the nodes that correspond to the actions taken by the agent $a$ and the most recent world observation $z$, and update the root, i.e., the node $h_{\mathfrak{r}}az$ becomes the new $\mathfrak{r}$, hence, $h_{\mathfrak{r}} = h_{\mathfrak{r}}az$, else;
• **Re-initialisation:** *if $T$ exists but, node $h_{\mathfrak{r}}a$ or $h_{\mathfrak{r}}az$ does not exist,* the tree is re-initialised. That is, we create a new node $(h, \mathcal{V}(h), \mathcal{N}(h), \mathcal{F}_h)$ with $h$ as the current history, $\mathcal{V}(h) = 0$, $\mathcal{N}(h) = 0$ and $\mathcal{F}_h$ is empty. Afterwards, we assign this new node as our new root $\mathfrak{r}$ of $\mathbf{T}$, which means that all other simulations already made in the agent's head are discarded and we restart the algorithm.

**In contrast to POMCP**, we *estimate the current probability $P(z|h_{\mathfrak{r}}a)$*, in this step, when finding the observation $z$ after taking the action $a$ considering our history $h_{\mathfrak{r}}$. Overall, we aim to use this probability to enhance the Particle Filter Update (see step (ii)) procedure by weighting its diversification and reinvigoration levels according to the agent's uncertainty when stepping into the new root – which would be the same as calculating the agent's "surprise" on finding the received observation in the real world after taking the chosen action. However, we consider that a compact representation of transition $\mathcal{T}$ or observation probabilities $\mathcal{Z}$ may not be available for complex problems. Hence, we propose the approximation of this probability using the knowledge within our particle filter $\mathcal{F}_{\mathfrak{r}}$ (without requiring a POMDP explicit model) by $P(z|h_{\mathfrak{r}}a) \approx \tilde{P}(z|h_{\mathfrak{r}}a) = \frac{\mathcal{N}(h_{\mathfrak{r}}az)}{\mathcal{N}(h_{\mathfrak{r}}a)}$,

where $\mathcal{N}(h_{\mathfrak{r}}az)$ and $\mathcal{N}(h_{\mathfrak{r}}a)$ are the number of visits of node $h_{\mathfrak{r}}az$ and $h_{\mathfrak{r}}a$, respectively (the new root node and its parent). If $h_{\mathfrak{r}}$ is empty or no new root node is found, $\tilde{P}(z|h_{\mathfrak{r}}a) = 0$. Note also that $\mathcal{N}(haz) \leq \mathcal{N}(ha), \forall h \in \mathbf{T}$ and, consequently, $\mathcal{N}(haz)/\mathcal{N}(ha) \in [0, 1]$.

**(ii) Particle Filter Initialisation and Update –** This process is responsible for initialising the particle filter of the root node $\mathcal{F}_{\mathfrak{r}}$ (if it is empty or does not exist) or performing the particle reinvigoration process based on the probability $\tilde{P}(z|h_{\mathfrak{r}}a)$ calculated in the Root Update process (step (i)). Directly, the initialisation is made through the sampling of $k$ particles (which generate the observation $z$) from the uniform distribution $\mathcal{U}_z$. On the other hand, the update considers the $\tilde{P}(z|h_{\mathfrak{r}}a)$ as the weight that balances the particle reinvigorating process over $\mathcal{F}_{\mathfrak{r}}$. The idea is to diversify (or boost) the new root's particle filter as a reliable approximation of the belief state function. If the probability of stepping into this new root is high, i.e., $\tilde{P}(z|h_{\mathfrak{r}}a)$ is high, we assume that the particles in the new root's particle filter will well approximate the real world, since through the simulations we recurrently found the observation $z$ after taking the action $a$ from the last root node $h$. In contrast, when this probability is low, we diversify the new root's particle filter $\mathcal{F}_{\mathfrak{r}}$ by uniformly generating particles that may better represent the real world using $\mathcal{U}_z$ instead of $\mathcal{F}_{\mathfrak{r}}$. Consequently, coherent with the above rationale, we reinvigorate the new root's particle filter $\mathcal{F}_{\mathfrak{r}}$ by maintaining $\lfloor k\tilde{P}(z|h_{\mathfrak{r}}a)\rfloor$ particles sampled from itself, i.e., $\mathcal{F}_{haz}$, and sampling new $\lceil k(1 - \tilde{P}(z|h_{\mathfrak{r}}a))\rceil$ particles from the uniform distribution $\mathcal{U}_z$. This update on the new root's particle filter may offer a better start for the algorithm when performing the search process since the sampling of belief states may consider an enhanced set of particles through our uncertainty-guided reinvigoration process. We kindly refer the reader to Appendix A for further discussion and pseudo-code to implement this procedure.

**(iii) Updating the Tree Exploration Coefficient –** To explain how we update our tree exploration coefficient, we first introduce the *(a) adaptation made to the traditional exploration coefficient*, then we present our *(b) modified entropy function*, which is used in the update, we explain how to calculate it and, in the end, we show our *(c) strategy to normalise the entropy* in the online planning context.

*(a) Exploration Coefficient Adaptation –* Before the actual tree simulation process, IB-POMCP first adjusts the value of our tree's exploration parameter based on the estimated entropy for the current system (the tree) in the root level $\mathfrak{r}$ with history $h_{\mathfrak{r}}$. Directly, our approach considers the replacement of the traditional $c$ constant by the function $(1 - \alpha(h_{\mathfrak{r}})) \in [0, 1]$, which we define as:

$$\alpha(h_{\mathfrak{r}}) := \frac{e \ln(\mathcal{N}(h_{\mathfrak{r}}))}{\mathcal{N}(h_{\mathfrak{r}})} \frac{\sum_{i=1}^{\mathcal{N}(h_{\mathfrak{r}})} \mathcal{H}_i(h_{\mathfrak{r}})}{\mathcal{N}(h_{\mathfrak{r}}) \max_{i=1}^{\mathcal{N}(h_{\mathfrak{r}})} \mathcal{H}_i(h_{\mathfrak{r}})} \tag{1}$$

Our insight is to use $\alpha$ to augment the current uncertainty level in the tree's search policy. On the left-hand side of the multiplication, we design a function that represents the chance of finding new information by exploring a node, which decreases as the number of visits to the node increases. $e$ is the Euler's constant, which will be used as our amortisation factor for $\ln(\mathcal{N}(h_{\mathfrak{r}}))/\mathcal{N}(h_{\mathfrak{r}})$. Applying both together, we can describe a function that slowly decreases and maintains, in the infinity and under some assumptions, theoretical guarantees for belief approximation (see Section 5). The right-hand side of the multiplication expresses the "*general surprise trend*", which is calculated through the division between the actual cumulative entropy $\sum_{i=1}^{\mathcal{N}(h_{\mathfrak{r}})} \mathcal{H}_i(h_{\mathfrak{r}})$ and the estimated maximum cumulative entropy $\mathcal{N}(h_{\mathfrak{r}}) \max_{i=1}^{\mathcal{N}(h_{\mathfrak{r}})} \mathcal{H}_i(h_{\mathfrak{r}})$, which is the multiplication of the total number of visits to the node and the maximum entropy. By multiplying both sides of the equation, we can estimate the current uncertainty of our system, in this case, of our tree. In addition to improving reasoning, we discard the need for prior knowledge about the problem of adjusting and fixing the tree constant $c$. Note that we adjust $\alpha(h_{\mathfrak{r}})$ for each traversal on the tree, i.e., from the root $\mathfrak{r}$ to some leaf.

*(b) Entropy Calculation –* We adapt Shannon's Entropy equation to measure the level of information in the IB-POMCP's search process based on the collection of observations of our agent while performing Simulations (see step (iv)), which is designed as $\mathcal{H}(h) = -\sum_{z\in\tilde{\mathbf{Z}}} P_h(z) \ln(P_h(z))$, where $P_h(z)$ is the probability of finding the observation $z \in \tilde{\mathbf{Z}}$ by simulating actions from the node with history $h$. We use $\tilde{\mathbf{Z}}$, which is an estimated set of observations, in the calculation of the entropy because we consider that a compact representation of the full observations space $\mathbf{Z}$ may not be available.

In order to build $\tilde{\mathbf{Z}}$ as a reliable estimation of $\mathbf{Z}$, we collect all the observations found during *all* traversals within the tree and save them first as the *multiset* $\tilde{\mathbf{Z}}^m$ – avoiding losing information, e.g., the frequency of the observations – and then we translate it as the *set* $\tilde{\mathbf{Z}}$ when necessary.

Each node has its own estimated multiset of observation $\tilde{\mathbf{Z}}_h^m$. Every time we visit a node $h$, we update the $\tilde{\mathbf{Z}}_h^m$ using the collected observation information, which is saved and back-propagated as $\tilde{\mathbf{Z}}_t^m$. On the other hand, $\tilde{\mathbf{Z}}_t^m$ is the multiset that saves the observations from a single traversal starting from the node at the tree level $t$ to the maximum length of the path $D$. Therefore, each node $h$ has its own multiset that saves all possible observations to be found by performing a simulation from it. Formally, we can define $\tilde{\mathbf{Z}}_h^m = \bigcup_{i=1}^{\mathcal{N}(h)} \tilde{\mathbf{Z}}_{t,i}^m$, where $\tilde{\mathbf{Z}}_{t,i}^m$ is the back-propagated $\tilde{\mathbf{Z}}_t^m$ at $i$-th visit to the node, and $\tilde{\mathbf{Z}}_{t,i}^m = \tilde{\mathbf{Z}}_{t+1,i}^m \cup z_t, \forall t = 0, 1, ..., D$. Note that $\tilde{\mathbf{Z}}_{t,i}^m = \emptyset, \forall t > D$. Let's put it as an example:

Consider $h = h_0$ as the root node, $D = 3$ and a single traversal in the tree. Under the back-propagation perspective, we start our update from the last node visited, related to $\tilde{\mathbf{Z}}_{t+3}^m$. Following our definition, $\tilde{\mathbf{Z}}_{t+3}^m = \{z_{t+3}\}$ and, since it is a leaf node, only $z_{t+3}$ will be included in $\tilde{\mathbf{Z}}_{h_3}^m$ as a new possible observation to be found from node $h_3$. Now, in $\tilde{\mathbf{Z}}_{t+2}^m$, we will add the $z_{t+2}$ to our back-propagation multiset $\tilde{\mathbf{Z}}_{t+3}^m$ and, consequently, we will add $\{z_{t+3}, z_{t+2}\}$ to $\tilde{\mathbf{Z}}_{h_2}^m$. We repeat this process until we reach $h_0$, where we include all found observations $\tilde{\mathbf{Z}}_t^m = \tilde{\mathbf{Z}}_{t+1}^m \cup z_0 = \{z_3, z_2, z_1, z_0\}$ to the root multiset $\tilde{\mathbf{Z}}_h^m$. Figure 1 illustrates this example and our proposed observation back-propagation procedure. Figures 1a and 1d present a high-level perspective of the process, showing only the observation nodes of the tree to facilitate visualisation. Figures 1b and 1c present a closer perspective of the process as a one-step execution. The red particles are states and the blue ones are observations.

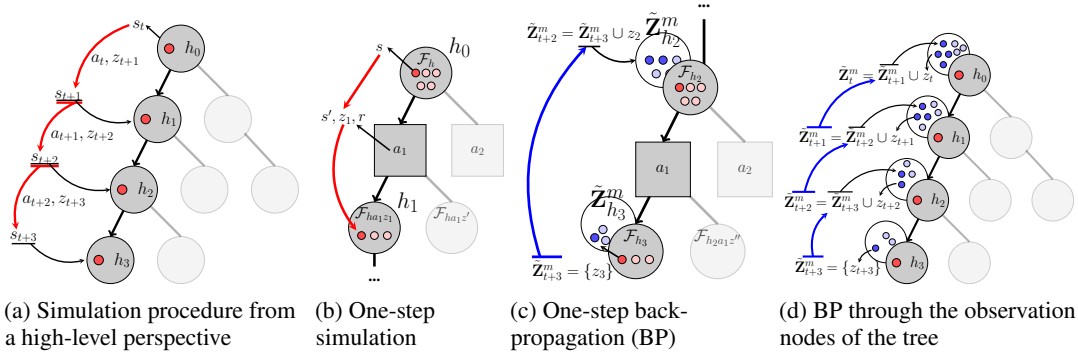

(a) Simulation procedure from a high-level perspective    (b) One-step simulation    (c) One-step back-propagation (BP)    (d) BP through the observation nodes of the tree

Figure 1: Illustration of IB-POMCP's search process.

With $\tilde{\mathbf{Z}}_h^m$ in hands, we now can approximate $P_h(z)$ by calculating the frequency of the observation $z$ in $\tilde{\mathbf{Z}}_h^m$, following $P_h(z) \approx \tilde{P}_h(z) = \frac{1}{|\tilde{\mathbf{Z}}_h^m|} \sum_{z' \in \tilde{\mathbf{z}}_h^m} 1_{\{z'=z\}}$. Therefore, our final entropy function will be $\mathcal{H}(h) = -\sum_{z \in \tilde{\mathbf{Z}}_h} \tilde{P}_h(z) \ln(\tilde{P}_h(z))$. We use this defined entropy to calculate the current entropy of the system. However, since IB-POMCP plans in an online manner and the $\tilde{\mathbf{Z}}_h^m$ is updated at each visit to the node, we propose the estimation of our tree's entropy using the update-equation:

$$\mathcal{H}_{\mathcal{N}(h)}(h) := \mathcal{H}_{\mathcal{N}(h)-1}(h) + \frac{(\mathcal{H}(h) - \mathcal{H}_{\mathcal{N}(h)-1}(h))}{\mathcal{N}(h)} \tag{2}$$

where $\mathcal{N}(h)$ is the number of visits to the node $h$, $\mathcal{H}(h)$ is the above defined entropy calculation, and $\mathcal{H}_i(h), i = 1, 2..., \mathcal{N}(h)$ is the estimated entropy at the visit $i$. Note that, by following this final definition for the entropy, we can also calculate $\alpha$ (Equation 1) and update it in an online fashion.

*(c) Entropy Normalisation* – Another issue that arises is that, since we may not have access to the true observation distribution in advance, our current approach will result in entropy calculations that are not standardised under the same image function. This problem occurs because the size of the set $\tilde{\mathbf{Z}}$ grows every time an observation never found before is delivered by the environment. As a consequence, the maximum value for the entropy also grows, and nodes that receive a higher number of different observations will calculate higher values for $\mathcal{H}(h)$, which can create bias by including these overestimated values in the decision-making process. Therefore, we propose its normalisation *a posteriori*, following: $\hat{\mathcal{H}}(h) := \frac{\mathcal{H}_{\mathcal{N}(h)}(h)}{\max_{j=1}^{\mathcal{N}(h)} \mathcal{H}_j(h)}$, where $\mathcal{H}_j(h)$ represents the entropy calculated at the $j$-th visit to the node $h$. This approach guarantees $\hat{\mathcal{H}}(h) \in [0, 1], \forall h$, i.e., for all nodes. Moreover,

$\mathcal{H}(h) = 1$ when $\mathcal{N}(h) = 0$ by definition, since we consider that when there is no information, the uncertainty is maximised, hence, the entropy is also maximised.

**(iv) Simulation –** After adjusting our $\alpha$ function, we start the simulation process. While expanding the tree, we propose a modified version of UCB1 to guide the tree search process based on the information level of each node, the *I-UCB* function, which follows:

$$\text{I-UCB}(h, a, \alpha) := \mathcal{V}(ha) + (1 - \alpha)\sqrt{\ln(\mathcal{N}(h))/\mathcal{N}(ha)} + \alpha\hat{\mathcal{H}}(ha) \tag{3}$$

Our proposal intends to guide the expansion and planning process based on entropy updates that occur within the tree, defining what we denote as the *information-guided planning process*. In this type of planning, a trade-off between exploration and exploitation still exists, but now we also bring information value to the discussion. Our main novelty here moves past the traditional idea, where the more we explore, the better we can exploit the rewards, thereby improving the agent's performance. Instead, we expand the discussion and enhance the algorithm capabilities by introducing an information-gain perspective to the tree-planning process using entropy calculations.

Intuitively, our action selection function considers that **(a)** if the system uncertainty is high (hence, $\alpha$ value is high), we weight the action selection based on the advantage of information gain (which may expand the tree in depth), trying to decrease the entropy value and accumulate knowledge, or; **(b)** if the system uncertainty is low (hence, $\alpha$ value is low), we weight the action selection in the advantage of the exploration gain (which may expand the tree in breadth), trying to increase the entropy value and increase the number of possibilities to reason over. Therefore, I-UCB performs adaptive planning that also depends on the system entropy, besides the upper confidence estimations.

Another direct advantage of the application of I-UCB instead of UCB1 is that our method rarely fails to deliver an action justified by metrics, that is, it rarely delivers a randomly chosen action as the best one. Usually, this issue arises in problems where rewards are sparsely distributed over a long horizon of actions, which can lead to several ties in the Q-values (e.g., at zero). Proposing a solution that works under these conditions requires **(a)** to increase the reasoning horizon of the solution, which usually leads to the expenditure of significantly more computational resources, or **(b)** the capability to handle it within the available reasoning horizon, which is a non-trivial task. IB-POMCP solves this problem by following a non-trivial solution considering **(a)** the inclusion of a novel metric for evaluation – the system entropy, which frequently is non-zero since the observation distribution is diverse and; **(b)** the adaptive value of alpha, which often promotes an action branch to find (at least) a non-zero value (reward or entropy) or to be frequently visited during the reasoning process.

**(v) Best action selection –** we decide the best action by picking the action with the highest score using $a_{best} = argmax_{a \in \mathbf{A}} (1 - \alpha)\mathcal{V}(ha) + \alpha\hat{\mathcal{H}}(ha)$. If there is a tie, we break it using the number of visits for each possible node. If it persists, we run a random policy to select the "best" action. Because of the proposed entropy calculation (and in contrast to POMCP), IB-POMCP has the ability to choose actions even without experiencing non-zero rewards while planning.

Lastly, considering the above discussion on adapting and inserting $\alpha$ inside I-UCB, we consider scaling $\alpha$ value to be within an interval $[q, 1 - q] \subset [0, 1], 0 < q < 0.5$, as a *practical* enhancement. This strategy allows our method to avoid ignoring part of the I-UCB result by multiplying one term by zero. This step will not be considered in the theoretical analysis.

**Algorithm/Pseudo-code:** To highlight the difference between POMCP and IB-POMCP, we present the complete pseudocode of our algorithm in Appendix A[1].

## 5 Theoretical Analysis

In this section, we analyse IB-POMCP's capability to plan optimally in partially observable finite horizon problems and $\epsilon$-optimally for infinite horizon partially observable problems. For the complete proofs see Appendix B. We start with the following assumption:

**Assumption 1** *Given a generic $POMDP$ represented by the tuple* $(\mathbf{S}, \mathbf{A}, \mathcal{R}, \mathcal{T}, \mathbf{Z}, \mathcal{Z}, \mathbf{H})$*, the $\mathcal{R}$ is bounded and there exists a positive maximum reward value* $r_{max} \in \mathcal{R}$*.*

---

[1]IB-POMCP's implementation publicly available on GitHub: https://github.com/lsmcolab/ib-pomcp/

Under this simple assumption and to establish the IB-POMCP's planning capabilities, we must examine our action selection procedure, specifically through the application of I-UCB. Consequently, we first analyse how the $\alpha$ coefficient behaves and impacts the IB-POMCP's search process:

**Lemma 1** $\alpha(h) \in [0, 1]$ *converges to 0 as the number of visits* $\mathcal{N}(h)$ *approach the infinity.*

**Theorem 1** *For any non-empty history* $h \neq \emptyset \in \mathbf{H}$ *and* $\forall a \in \mathbf{A}$, *as* $\mathcal{N}(h) \to \infty$, $\mathcal{N}(ha) \to \infty$, *we have that all states* $b$ *in* $\mathcal{B}(h, b)$ *which* $P(s = b \mid h) > 0$ *will be simulated infinitely many times.*

Using the previous results, we can show the convergence for a finite horizon:

**Theorem 2** *IB-POMCP's nodes converge to the optimal value for a fixed horizon* $D$ *as the number of search iterations goes to infinite (i.e.,* $\mathcal{N}(h_{\mathfrak{r}}) \to \infty$).

We now move on to the infinite horizon case. Consider $\mathcal{V}^*(h_{\mathfrak{r}})$ as the optimal value function for the root $\mathfrak{r}$ and $\mathcal{V}_D^*(h_{\mathfrak{r}})$ as the optimal value function for the root $\mathfrak{r}$ for a finite horizon D. We find that:

**Theorem 3** *Given any* $\epsilon$, *there is a finite horizon* $D$ *for the problem, such that* $|\mathcal{V}^*(h_{\mathfrak{r}}) - \mathcal{V}_D^*(h_{\mathfrak{r}})| \leq \epsilon$, *where* $h_{\mathfrak{r}}$ *represents the root of the tree.*

Given the previous results, we can prove the following for the convergence in the infinite horizon:

**Corollary 1** *IB-POMCP converges to* $\epsilon$*-optimal actions at the root* $(h_{\mathfrak{r}})$ *in a* $\gamma$*-discounted infinite horizon scenario.*

## 6 Results

**Evaluation Settings –** We define five well-known domains as our benchmarks. See Appendix C for more information. **(i) The Tiger (T0)** environment presents an agent *must choose between two doors: one hides a treasure; the other, a tiger.* The agent can decide which door to open or listen to before making the final decision. **(ii) The Maze (M0-M3)** environment [36], which is an *Active Localisation Problem* in which an agent navigates through a toroidal grid and *tries to localise its own position* by collecting local information. The reward is based on the entropy of the current belief. **(iii) The RockSample (R0-R3)** problem [36] considers that a robot has to navigate through a grid-world and *maximise the number of good rocks it picks up and analyses instead of bad rocks*. If it picks up (samples) a bad rock, it gets a negative reward. Otherwise, it gets a positive reward and the good rock becomes bad. **(iv) The Tag/LaserTag (LT0-LT1)** environment [34], where *an agent is trying to find and tag a target opponent agent that intentionally moves away*. The agent only knows its own position but can observe the target's position if they are in the same position (Tag) or using an 8-directional laser (in LaserTag). If the agent successfully tags the target it is rewarded, otherwise it is penalised. **(v) The Foraging (F0-F4)** problem is a famous scenario to evaluate online planning algorithms [2, 30] that presents *an agent that collects items displaced in a rectangular grid world*. All experiments were implemented using *AdLeap-MAS* [9] because it implements all five environments and enables the quick execution of our experiments. Each run was performed in a single node of a high-performance cluster containing 16 cores of Intel Ivy Bridge processors and 64 GB RAM.

**Baselines –** In this work, we compare IB-POMCP against: **(a) POMCP** [32], since it is a relevant state-of-art proposal and represents the basis of this work; **(b)** $\rho$**-POMCP** [36], representing our main competitor in terms of using information theory to perform the online planning procedure, and **(c) TB** $\rho$**-POMCP** [36], as a faster alternative to $\rho$-POMCP that employs an explicit POMDP model.

**Metrics –** Two different evaluation metrics were used: **(i) the average reward (R)** across the experiment's mean reward; and **(ii) the average planning time (t)** spent by the agent to plan the next actions. Mean results were calculated across *50 executions*. Every experiment ran independently; thus, no knowledge was carried from one execution to another. The calculated errors ($\pm Err$) represent the $95\%$ confidence interval of a two-sample t-test. Note that our results and baselines are separated into two categories **(i) Quick Planning**: grouping methods that perform the decision-making process within a reasonable time window; and **(ii) Long Planning**: presenting the $\rho$-POMCP's results (without time constraints). We separated them to make their understanding easier.

Finally, we refer the reader to our Appendix D to information about our hyperparameters set.

**Benchmarks study** – All results are presented in Table 1. In addition, we highlight the pros and cons of our method, analysing our limitations while discussing the outcome for each environment.

Table 1: Summarized results. The bold underlined values in *Quick Planning* highlight the best result with statistical significance across all *Quick Planning* baselines for the respective metric.

| Problem | Quick Planning | | | | | | Long Planning | |
|---|---|---|---|---|---|---|---|---|
| | **POMCP** | | **TB $\rho$-POMCP** | | **IB-POMCP** | | **$\rho$-POMCP** | |
| | $R \pm Err$ | $t \pm Err$ (sec) | $R \pm Err$ | $t \pm Err$ (sec) | $R \pm Err$ | $t \pm Err$ (sec) | $R \pm Err$ | $t \pm Err$ (sec) |
| Tiger (T0) | $-4.25 \times 10^2$ $\pm 0.80 \times 10^2$ | $\underline{\mathbf{0.09 \pm 0.01}}$ | $\underline{\mathbf{-0.16 \times 10^2}}$ $\underline{\mathbf{\pm 0.03 \times 10^2}}$ | $0.20 \pm 0.00$ | $-0.52 \times 10^2$ $\pm 0.15 \times 10^2$ | $0.11 \pm 0.01$ | $-0.21 \times 10^2$ $\pm 0.03 \times 10^2$ | $1.45 \pm 0.04$ |
| MazeCross (M0) | $1.11 \times 10^{-2}$ $\pm 0.08 \times 10^{-2}$ | $2.47 \pm 0.02$ | $1.12 \times 10^{-2}$ $\pm 0.07 \times 10^{-2}$ | $3.00 \pm 0.00$ | $\underline{\mathbf{1.23 \times 10^{-2}}}$ $\underline{\mathbf{\pm 0.04 \times 10^{-2}}}$ | $\mathbf{2.43 \pm 0.01}$ | $1.19 \times 10^{-2}$ $\pm 0.07 \times 10^{-2}$ | $28.16 \pm 0.68$ |
| MazeHoles (M1) | $1.56 \times 10^{-2}$ $\pm 0.50 \times 10^{-2}$ | $3.72 \pm 0.03$ | $1.32 \times 10^{-2}$ $\pm 0.46 \times 10^{-2}$ | $4.00 \pm 0.00$ | $\underline{\mathbf{3.80 \times 10^{-2}}}$ $\underline{\mathbf{\pm 0.86 \times 10^{-2}}}$ | $\mathbf{3.60 \pm 0.04}$ | $2.34 \times 10^{-2}$ $\pm 0.44 \times 10^{-2}$ | $48.41 \pm 2.82$ |
| MazeDots (M2) | $1.20 \times 10^{-2}$ $\pm 0.28 \times 10^{-2}$ | $2.59 \pm 0.02$ | $0.79 \times 10^{-2}$ $\pm 0.15 \times 10^{-2}$ | $3.00 \pm 0.00$ | $\underline{\mathbf{2.99 \times 10^{-2}}}$ $\underline{\mathbf{\pm 0.85 \times 10^{-2}}}$ | $\mathbf{2.53 \pm 0.02}$ | $1.25 \times 10^{-2}$ $\pm 0.20 \times 10^{-2}$ | $36.71 \pm 1.39$ |
| MazeGridX (M3) | $\underline{\mathbf{2.80 \times 10^{-2}}}$ $\underline{\mathbf{\pm 0.60 \times 10^{-2}}}$ | $1.86 \pm 0.02$ | $0.99 \times 10^{-2}$ $\pm 0.15 \times 10^{-2}$ | $2.00 \pm 0.00$ | $0.96 \times 10^{-2}$ $\pm 0.02 \times 10^{-2}$ | $\mathbf{1.82 \pm 0.01}$ | $1.37 \times 10^{-2}$ $\pm 0.30 \times 10^{-2}$ | $19.79 \pm 0.32$ |
| RockSample22 (R0) | $0.06 \times 10^{-1}$ $\pm 0.01 \times 10^{-1}$ | $\underline{\mathbf{2.08 \pm 0.03}}$ | $0.03 \times 10^{-1}$ $\pm 0.01 \times 10^{-1}$ | $5.00 \pm 0.00$ | $0.07 \times 10^{-1}$ $\pm 0.01 \times 10^{-1}$ | $2.33 \pm 0.04$ | $0.03 \times 10^{-1}$ $\pm 0.01 \times 10^{-1}$ | $2.70 \pm 0.43$ |
| RockSample40 (R1) | $0.12 \times 10^{-1}$ $\pm 0.01 \times 10^{-1}$ | $\underline{\mathbf{2.08 \pm 0.03}}$ | $0.11 \times 10^{-1}$ $\pm 0.02 \times 10^{-1}$ | $5.00 \pm 0.00$ | $\underline{\mathbf{0.16 \times 10^{-1}}}$ $\underline{\mathbf{\pm 0.01 \times 10^{-1}}}$ | $2.40 \pm 0.04$ | $0.10 \times 10^{-1}$ $\pm 0.01 \times 10^{-1}$ | $2.77 \pm 0.32$ |
| RockSample44 (R2) | $0.07 \times 10^{-1}$ $\pm 0.01 \times 10^{-1}$ | $\underline{\mathbf{4.67 \pm 0.09}}$ | $0.06 \times 10^{-1}$ $\pm 0.01 \times 10^{-1}$ | $5.00 \pm 0.00$ | $\underline{\mathbf{0.09 \times 10^{-1}}}$ $\underline{\mathbf{\pm 0.02 \times 10^{-1}}}$ | $5.10 \pm 0.06$ | $0.04 \times 10^{-1}$ $\pm 0.02 \times 10^{-1}$ | $23.18 \pm 3.10$ |
| RockSample17 (R3) | $-0.04 \times 10^{-1}$ $\pm 0.02 \times 10^{-1}$ | $\underline{\mathbf{4.72 \pm 0.09}}$ | $-0.06 \times 10^{-1}$ $\pm 0.02 \times 10^{-1}$ | $5.00 \pm 0.00$ | $\underline{\mathbf{0.00 \times 10^{-1}}}$ $\underline{\mathbf{\pm 0.01 \times 10^{-1}}}$ | $5.11 \pm 0.06$ | $-0.09 \times 10^{-1}$ $\pm 0.02 \times 10^{-1}$ | $25.49 \pm 3.28$ |
| Tag (LT0) | $-0.47 \times 10^{-1}$ $\pm 0.09 \times 10^{-1}$ | $1.22 \pm 0.20$ | $-0.65 \times 10^{-1}$ $\pm 0.09 \times 10^{-1}$ | $3.00 \pm 0.00$ | $\underline{\mathbf{-0.36 \times 10^{-1}}}$ $\underline{\mathbf{\pm 0.07 \times 10^{-1}}}$ | $\underline{\mathbf{0.86 \pm 0.14}}$ | $-0.65 \times 10^{-1}$ $\pm 0.10 \times 10^{-1}$ | $11.23 \pm 1.52$ |
| LaserTag (LT1) | $-0.90 \times 10^{-1}$ $\pm 0.06 \times 10^{-1}$ | $4.72 \pm 0.09$ | $-0.86 \times 10^{-1}$ $\pm 0.09 \times 10^{-1}$ | $3.00 \pm 0.00$ | $-0.92 \times 10^{-1}$ $\pm 0.06 \times 10^{-1}$ | $5.11 \pm 0.06$ | $-0.96 \times 10^{-1}$ $\pm 0.51 \times 10^{-1}$ | $19.41 \pm 1.02$ |
| TheCorridor (F0) | $4.29 \times 10^{-2}$ $\pm 0.51 \times 10^{-2}$ | $0.96 \pm 0.06$ | $4.36 \times 10^{-2}$ $\pm 0.38 \times 10^{-2}$ | $1.00 \pm 0.00$ | $\underline{\mathbf{6.89 \times 10^{-2}}}$ $\underline{\mathbf{\pm 0.28 \times 10^{-2}}}$ | $0.87 \pm 0.10$ | $5.15 \times 10^{-2}$ $\pm 0.18 \times 10^{-2}$ | $8.18 \pm 0.71$ |
| U-shaped (F1) | $0.70 \times 10^{-2}$ $\pm 0.34 \times 10^{-2}$ | $3.67 \pm 0.02$ | $0.53 \times 10^{-2}$ $\pm 0.03 \times 10^{-2}$ | $3.00 \pm 0.00$ | $\underline{\mathbf{5.10 \times 10^{-2}}}$ $\underline{\mathbf{\pm 0.14 \times 10^{-2}}}$ | $\underline{\mathbf{2.99 \pm 0.19}}$ | $5.30 \times 10^{-2}$ $\pm 0.31 \times 10^{-2}$ | $37.36 \pm 1.29$ |
| U-obstacles (F2) | $1.17 \times 10^{-2}$ $\pm 0.34 \times 10^{-2}$ | $1.90 \pm 0.03$ | $1.67 \times 10^{-2}$ $\pm 0.23 \times 10^{-2}$ | $2.00 \pm 0.00$ | $\underline{\mathbf{5.50 \times 10^{-2}}}$ $\underline{\mathbf{\pm 0.34 \times 10^{-2}}}$ | $1.88 \pm 0.08$ | $3.75 \times 10^{-2}$ $\pm 0.45 \times 10^{-2}$ | $15.44 \pm 0.37$ |
| Warehouse (F3) | $5.20 \times 10^{-2}$ $\pm 0.45 \times 10^{-2}$ | $3.02 \pm 0.11$ | $4.42 \times 10^{-2}$ $\pm 1.03 \times 10^{-2}$ | $3.00 \pm 0.00$ | $\underline{\mathbf{10.73 \times 10^{-2}}}$ $\underline{\mathbf{\pm 0.41 \times 10^{-2}}}$ | $2.91 \pm 0.25$ | $8.99 \times 10^{-2}$ $\pm 0.96 \times 10^{-2}$ | $20.40 \pm 0.67$ |
| TheOffice (F4) | $0.43 \times 10^{-2}$ $\pm 0.21 \times 10^{-2}$ | $1.90 \pm 0.03$ | $0.72 \times 10^{-2}$ $\pm 0.14 \times 10^{-2}$ | $2.00 \pm 0.00$ | $\underline{\mathbf{2.34 \times 10^{-2}}}$ $\underline{\mathbf{\pm 0.27 \times 10^{-2}}}$ | $1.88 \pm 0.08$ | $1.61 \times 10^{-2}$ $\pm 0.17 \times 10^{-2}$ | $15.44 \pm 0.37$ |

In *the Tiger domain*, TB $\rho$-POMCP presents the best average reward among the Quick Planning methods ($\rho < 0.01$). IB-POMCP still outperforms POMCP, showing significant improvement in terms of reward ($\rho < 0.01$). For this specific problem, since the action "listen" generates variation in the entropy, IB-POMCP keeps performing it repeatedly until other action value outcomes the "listen" value in the best action selection procedure (Section 4, step (v)), a circumstance which leads our method to reduce its average reward collection. Hence, when facing problems where seeking spots of high uncertainty produces small penalties, IB-POMCP may collect penalties until it significantly decreases the uncertainty and chooses another path.

In *the Maze domain*, we observe that IB-POMCP presents a significantly better average reward in 3 out of 4 proposed scenarios ($\rho \leq 0.015$), except for M3, for which POMCP presents a better result. Note that reducing uncertainty leads to increasing reward; i.e. the faster an agent can access areas with high uncertainty, the higher its received reward. IB-POMCP's results match our expectations (given the developed rationale throughout our methodology) since we build it to, besides tracking the rewards available in the scenario, often seek paths that lead to spots with high entropy in order to decrease uncertainty, what increases reward in this scenario. In terms of time, we present significantly faster reasoning in all scenarios ($\rho < 0.01$). Investigating why IB-POMCP runs faster than POMCP, we found that our information-guided planning leads the algorithm to perform more transitions during the rollout phase than while performing the simulation inside our actual search tree, with a rate $\frac{\#rollout}{\#simulation} = 1.61$, whereas POMCP presents a rate of $1.28$. Because rollout transitions run faster than simulation transitions inside the tree, we can save time by performing them frequently.

In *the Rock Sample problem*, we can see that IB-POMCP shows a significant improvement in terms of reward collection ($\rho \leq 0.02$) in 3 out of 4 scenarios – except for the simplest scenario RockSample22 (R0), which presents a p-value of $\rho \leq 0.11$. In terms of reasoning time, POMCP is slightly faster than all Quick Planning methods in 3 out of 4 scenarios ($\rho \leq 0.01$), except for RockSample17 (R3).

In *Tag*, IB-POMCP presents significant improvement in terms of reward collection and reasoning time against all baselines ($\rho \leq 0.04$). As in the Maze, we believe that IB-POMCP is faster because it simulates transitions in rollouts more often (in a ratio of $\frac{\#rollout}{\#simulation} = 1.7$ against $1.61$ for POMCP).

In *LaserTag*, we have a tie between all methods, with no statistically significant difference spotted across metrics ($\rho \geq 0.32$ for reward collection and $\rho \geq 0.06$ for reasoning time).

Finally, in the *Foraging domain*, our proposed method significantly outperformed all the baselines ($\rho < 0.01$). The Foraging problem represents our most complex scenario, and empirically shows how IB-POMCP can overcome the necessity of adjusting the reasoning horizon to perform planning in settings that deliver rewards sparsely. The only reward available in these scenarios comes from the collection of tasks. Consequently, while attempting to solve a task, the number of actions that the agent needs to plan and execute in sequence may approach or exceed the reasoning horizon size. In this case, the probability of experiencing this reward is low, and algorithms that only follow the reward in planning will rarely obtain it in their simulations; hence, they fail to plan effectively.

Overall, we experimentally demonstrated that our novel proposed method can significantly improve the performance of our agent and its planning capabilities without penalising the reasoning time. We also demonstrated that our proposal can improve the decision-making process by using only the information generated during the tree search process and, in contrast to $\rho$-POMCP, without relying on the explicit representation of latent functions (e.g., the observation function).

**Ablation study –** To evaluate the impact of our proposed modifications and enhancements, we performed an ablation study over our method. We consider 2 different variations of IB-POMCP that partially implement its key points in this experiment: **(a) the Information-guided Particle Reinvigoration POMCP (IPR-POMCP)**, which implements the modifications to the particle filter reinvigoration process (explained in Section 4, steps (i) and (ii)); and **(b) the I-UCB POMCP,** which implements the proposed modifications to the search and simulation process (explained in Section 4, steps (iii) and (iv)). We consider 4 different scenarios, one for each environment presented in Table 1, to run our experiments. The results are depicted in Figure 2. Both additional methods proposed for the study are available in our GitHub repository together with the complete code of this work.

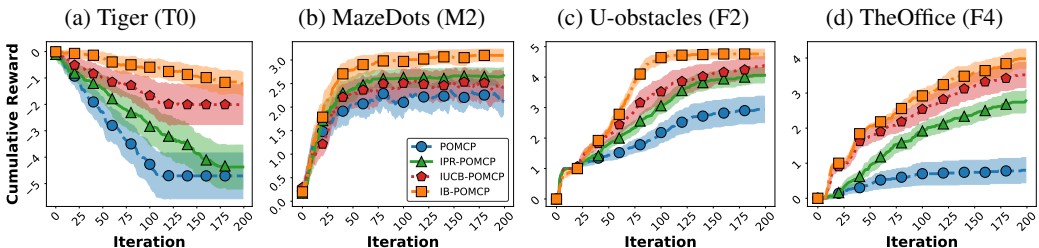

Figure 2: Ablation study of IB-POMCP in four different scenarios.

By analysing the graphs, it is clear that the I-UCB and implementation of an information-guided particle reinvigoration process for the Tiger, Maze, and Foraging problems directly enhance the reasoning capabilities of our planning method, which is translated here in terms of reward collection. For the Tiger and Foraging problems, diversifying the set of particles (I-UCB POMCP) alone presents improvements for POMCP; however, it has less impact than including entropy in the search process (IPR-POMCP). On the other hand, in the Maze environment, both approaches present similar improvements for the POMCP algorithm. However, when combined, they significantly improved upon the baseline results. Our intuition behind these results is that the IPR-POMCP proposal is responsible for delivering a better set of particles and a better initial estimation of the current state to the agent before simulating actions, whereas I-UCB continually guides and affects the planning process from the beginning of the simulations until the decision of the best action.

## 7  Conclusion

In this study, we propose *IB-POMCP*, a novel algorithm for planning under uncertainty that is capable of aggregating information entropy into a decision-making algorithm using our modified version of the UCB function, *I-UCB*. We present the theoretical properties for convergence under certain assumptions, which are supported by empirical results collected in five different domains. Overall, we increase the reward collection by up to 10 times in comparison with TB $\rho$-POMCP (U-shaped, F1), in addition to reducing the reasoning time by up to 93% compared to $\rho$-POMCP (MazeHoles).

## Acknowledgments and Disclosure of Funding

We gratefully acknowledge the financial support provided by a PhD studentship from the Faculty of Science and Technology (FST) at Lancaster University, United Kingdom. We also thank FST for granting us access to their High-End Computing Cluster (HEC 3.0), which was pivotal to this research. We extend our sincere gratitude to Dr Mike Pacey for his expertise and assistance provided to correctly setting up HEC 3.0 and running our experiments. Our special thanks go to Dr Ronghui Mu for providing insightful comments and feedback on an earlier draft of this paper.

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

# A    Algorithms

In this section, we provide the pseudo-code of the particle filter update step (Algorithm 1), and for IB-POMCP's main routine (Algorithm 2). To clarify our approach better and support the text presented in the main paper, both algorithms highlight the differences between our approach and POMCP [32]. See Section 3 in the main paper for further details about our rationale and methodology.

---

**Algorithm 1 Particle Filter Update.** The starred lines (with a light grey background) highlight the differences between POMCP and our proposal, IB-POMCP. *Union considers repetition here.*

---

1: **procedure** PARTICLEFILTERUPDATE$(\mathcal{F}_{\mathfrak{r}}, \tilde{P}(z|ha), k)$
2: $\quad$ $\hat{\mathbf{B}} \leftarrow \mathcal{F}_{\mathfrak{r}}, \mathcal{F}_{\mathfrak{r}} \leftarrow \{\}$ $\qquad\qquad\qquad\qquad\qquad\qquad$ ▷ $\hat{\mathbf{B}}$ is a copy of $\mathcal{F}_{\mathfrak{r}}$
*3: $\quad$ **while** $|\mathcal{F}_{\mathfrak{r}}| < \tilde{P}(z|ha)k$ **do**
*4: $\quad\quad$ $b \sim \hat{\mathbf{B}}, \mathcal{F}_{\mathfrak{r}} \leftarrow \mathcal{F}_{\mathfrak{r}} \cup b$ $\qquad\qquad$ ▷ Adding $\tilde{P}(z|ha)k$ particles to the new $\mathcal{F}_{\mathfrak{r}}$
*5: $\quad$ **while** $|\mathcal{F}_{\mathfrak{r}}| < k$ **do**
*6: $\quad\quad$ $b \sim \mathcal{U}_z, \mathcal{F}_{\mathfrak{r}} \leftarrow \mathcal{F}_{\mathfrak{r}} \cup b$ $\qquad\qquad$ ▷ Adding $(1 - \tilde{P}(z|ha))k$ particles to the new $\mathcal{F}_{\mathfrak{r}}$
*7: $\quad$ **return** $\mathcal{F}_{\mathfrak{r}}$

---

**Algorithm 1 Particle Filter Update – Details and Further Discussion:**    We save some space here with the intention of assuring the reader's understanding of the novelty that lies on our particle filter update proposal. Our method employs a dynamic threshold derived from the dynamic value of $\tilde{P}(z|ha)$ (or specifically, $1 - \tilde{P}(z|ha)$ that determines the ratio of uniform samples in the reinvigorated particle filter). Unlike some traditional approaches which use a fixed value for that threshold, or some other state-of-the-art contributions, which mostly rely on additional knowledge and expensive calculations [17, 14, 10], IB-POMCP does not need an explicit true distribution of observations, transition function or training data in this process by using our proposed $\tilde{P}(z|h_{\mathfrak{r}}a)$ probability.

It is important to note that the action selection process within the planning phase has a direct impact on the $\tilde{P}(z|h_{\mathfrak{r}}a)$ value. Therefore, the strategy used to choose actions for simulation during planning will affect the particle filter, since it affects the dynamic threshold $\tilde{P}(z|h_{\mathfrak{r}}a)$. Hence, note that the I-UCB equation used in the search tree also has an impact on belief tracking.

Our suggested algorithm to investigate the impact of this proposal (Ablation study in Section 6), IPR-POMCP, employs our dynamic threshold and shows significant improvement to the conventional POMCP, which follows a standard particle reinvigoration paradigm (with a fixed parameter for reinvigoration). Hence, just using the dynamic threshold $\tilde{P}(z|h_{\mathfrak{r}}a)$ improves results. Furthermore, the ablation study indicates that combining the dynamic threshold with the I-UCB equation yields even more favourable results and that using solely the I-UCB equation without the dynamic threshold (IUCB-POMCP baseline) results in worse outcomes than the full IB-POMCP approach. Hence, both the dynamic threshold and the I-UCB equation are important to improve results, and their simultaneous application produces even greater enhancements than when employed individually.

What is perhaps still left open is whether the I-UCB equation leads to better $\tilde{P}(z|h_{\mathfrak{r}}a)$ values than the UCB equation. Since our approach takes the estimated entropy into consideration for action selection we believe that I-UCB may lead to better threshold values.

For example, in cases where a node possesses high entropy and no reward in the reasoning horizon, the I-UCB equation could prioritise visiting it more often than the UCB equation would, increasing the $\tilde{P}(z|h_{\mathfrak{r}}a)$ value for that particular node (note that it will not imply in worse probability estimation because the entropy is based on the estimation of observations and the particle filter gives the probabilities of states). The high entropy indicates that the node accommodates a more diverse set of particles in the particle filter and hence sampling particles from a uniform distribution during particle reinvigoration (to increase diversity) becomes less useful. Hence, the implications of the I-UCB equation appear favourable as it leads to fewer particle samples from the uniform distribution in this case (as we use the ratio $1 - \tilde{P}(z|h_{\mathfrak{r}}a)$). On the other hand, if a node has low entropy, the particle filter may easily get biased to a small set of states. In this case, I-UCB would lead to a higher number of particles sampled from the uniform distribution than the UCB equation in the particle reinvigoration,

since it would have a lower $\tilde{P}(z|h_\mathfrak{r}a)$ value in I-UCB. Therefore, we believe that I-UCB should be better than UCB for defining the dynamic threshold $\tilde{P}(z|h_\mathfrak{r}a)$, and hence better in belief tracking.

---

**Algorithm 2 IB-POMCP's Planning.** The starred lines in a light grey background highlight the difference between POMCP and our proposal. We suggest the reader to Silver and Veness (2010) [32] for further details. $\gamma$ is the historical discount factor, $depth_{max}$ is the maximum depth for the tree, and $z_h$ represents the last observation found (associated with node $h$).

---

1: **procedure** SEARCH($h_\mathfrak{r}$)
2:     **while** $Timeout()$ **is** $False$ **do**
3:         **if** $\mathcal{F}_\mathfrak{r} = \emptyset$ **then** $s \sim \mathcal{U}_z$
4:         **else** $s \sim \mathcal{F}_\mathfrak{r}$
*5:             $\alpha := \dfrac{e \ln(\mathcal{N}(h_\mathfrak{r}))}{\mathcal{N}(h_\mathfrak{r})} \dfrac{\sum_{i=1}^{\mathcal{N}(h_\mathfrak{r})} \mathcal{H}_i(h_\mathfrak{r})}{\mathcal{N}(h_\mathfrak{r}) \max\limits_{i=1}^{\mathcal{N}(h_\mathfrak{r})} \mathcal{H}_i(h_\mathfrak{r})}$
*6:         $Simulate(s, h_\mathfrak{r}, 0, \alpha)$
*7:         **return** $\arg\max\limits_{a \in \mathbf{A}} (1-\alpha)\mathcal{V}(ha) + \alpha\hat{\mathcal{H}}(ha)$

1: **procedure** EXPAND_NODE($h$)
2:     **for** $a \in \mathbf{A}$ **do**
3:         $\mathbf{T}(ha) \leftarrow (ha, \mathcal{V}_{init}(ha), \mathcal{N}_{init}(ha), \emptyset)$

1: **procedure** ROLLOUT($s, h, d, \gamma$)
2:     **if** $d < depth_{max}$ **then return** $0$
3:     $a \sim \pi_{rollout}(h, \cdot)$
4:     $(s', z, r) \sim \mathcal{G}(s, a)$
5:     **return** $r + \gamma Rollout(s', haz, d+1)$

1: **procedure** SIMULATE($s, h, d, \alpha$)
2:     **if** $d < depth_{max}$ **then**
*3:         **return** $0, \{z_h\}$
4:     **if** $h \notin \mathbf{T}$ **then**          ▷ If node is not in the tree
5:         $Expand\_Node(h)$
*6:         **return** $Rollout(s, h, d), \{z\}$
*7:     $a \leftarrow \arg\max\limits_{a \in \mathbf{A}}$ I-UCB$(h, a, \alpha)$
8:     $(s', z, r) \sim \mathcal{G}(s, a)$
*9:     $r', \tilde{\mathbf{Z}}_{t+1}^m \leftarrow Simulate(s', haz, d+1, \alpha)$
10:    $\mathcal{F}_h \leftarrow \mathcal{F}_h \cup \{s\}$
11:    $\mathcal{N}(h) \leftarrow \mathcal{N}(h) + 1$
12:    $\mathcal{N}(ha) \leftarrow \mathcal{N}(ha) + 1$
13:    $\mathcal{V}(ha) \leftarrow \mathcal{V}(ha) + \dfrac{R - \mathcal{V}(ha)}{\mathcal{N}(ha)}$
*14:    $\mathcal{H}_i(h) := \mathcal{H}_{i-1}(h) + \dfrac{(\mathcal{H}_i(h) - \mathcal{H}_{i-1}(h))}{\mathcal{N}(h)}$
*15:    $R \leftarrow r + \gamma r'$
*16:    $\tilde{\mathbf{Z}}_h^m \leftarrow \tilde{\mathbf{Z}}_h^m \cup \tilde{\mathbf{Z}}_{t+1}^m$
*17:    **return** $R, \tilde{\mathbf{Z}}_{t+1}^m \cup z$

---

# B Detailed Theoretical Analysis

In this section, we offer a comprehensive theoretical analysis of our proposed method. In addition to the assumption, lemma, theorems, and corollary presented in the main paper, our complete analysis encompasses all proofs, along with our definition (Definition 1) and lemma (Lemma 2).

**Definition 1** An estimated $\hat{\mathcal{V}}(s)$ is said to be $\epsilon-$optimal if $|\hat{\mathcal{V}}(s) - \mathcal{V}^*(s)| \leq \epsilon$. From the literature, actions taken from the result of $\epsilon$-optimal value function estimations are known as $\epsilon$-optimal actions.

**Assumption 2** *Given a generic POMDP represented by the tuple* $(\mathbf{S}, \mathbf{A}, \mathcal{R}, \mathcal{T}, \mathbf{Z}, \mathcal{Z}, \mathbf{H})$, *the* $\mathcal{R}$ *is bounded and there exists a positive maximum reward value* $r_{max} \in \mathcal{R}$.

**Lemma 1** $\alpha(h) \in [0, 1]$ *converges to 0 as the number of visits* $\mathcal{N}(h)$ *approach the infinity. Mathematically,* $\lim_{\mathcal{N}(h) \to \infty} \alpha(h) = 0$.

**Proof:** Considers our equation for a root node with history $h$:

$$\alpha(h) = \frac{e \ln(\mathcal{N}(h))}{\mathcal{N}(h)} \frac{\sum_{i=1}^{\mathcal{N}(h)} \mathcal{H}_i(h)}{\mathcal{N}(h) \max\limits_{i=1}^{\mathcal{N}(h)} \mathcal{H}_i(h)}$$

First, we can re-organise the equation as follows:

$$\alpha(h) = \frac{e \ln(\mathcal{N}(h))}{\mathcal{N}(h)} \frac{\sum_{i=1}^{\mathcal{N}(h)} \mathcal{H}_i(h)}{\mathcal{N}(h) \max_{i=1}^{\mathcal{N}(h)} \mathcal{H}_i(h)} = \frac{e \ln(\mathcal{N}(h))}{\mathcal{N}(h)^2} \frac{\sum_{i=1}^{\mathcal{N}(h)} \mathcal{H}_i(h)}{\max_{i=1}^{\mathcal{N}(h)} \mathcal{H}_i(h)}$$

Analysing the limit over the new expression, we can find that:

$$\lim_{\mathcal{N}(h)\to\infty} \frac{e\ln(\mathcal{N}(h))}{\mathcal{N}(h)^2} \frac{\sum_{i=1}^{\mathcal{N}(h)} \mathcal{H}_i(h)}{\max_{i=1}^{\mathcal{N}(h)} \mathcal{H}_i(h)} = 0$$

by following the rationale:

$$0 \le \sum_{i=1}^{\mathcal{N}(h)} \mathcal{H}_i(h) \le \mathcal{N}(h) \max_{i=1}^{\mathcal{N}(h)} \mathcal{H}_i(h) \Rightarrow 0 \le \alpha(h) \le \frac{e\ln(\mathcal{N}(h))}{\mathcal{N}(h)^2} \frac{\mathcal{N}(h) \max_{i=1}^{\mathcal{N}(h)} \mathcal{H}_i(h)}{\max_{i=1}^{\mathcal{N}(h)} \mathcal{H}_i(h)}$$

Therefore, since we know that:

$$\lim_{\mathcal{N}(h))\to\infty} \frac{eln(\mathcal{N}(h))}{\mathcal{N}(h)^2} \frac{\mathcal{N}(h) \max_{i=1}^{\mathcal{N}(h)} \mathcal{H}_i(h)}{\max_{i=1}^{\mathcal{N}(h)} \mathcal{H}_i(h)} =$$

$$\frac{eln(\mathcal{N}(h))}{\mathcal{N}(h)^2}\mathcal{N}(h) = e\frac{\mathcal{N}(h)ln(\mathcal{N}(h))}{\mathcal{N}(h)^2} = 0, \text{ by L'Hopital's Rule}$$

using the sandwich theorem, we find that $\alpha(h) \to 0$ as $\mathcal{N}(h) \to \infty$ ■

Now, we introduce Lemma 2, which will be used as a key result that supports Theorem 1. In other words, we will use Lemma 2's proof to reach a contradiction later in order to prove Theorem 1.

**Lemma 2** *Assume an action $a_i$ that is taken a finite number of times and $a_j$ that is taken infinitely during IB-POMCP's search for any node $h$. There $\exists t'' > t'$ such that $I\text{-}UCB_{t''}(ha_i) \ge I\text{-}UCB_{t''}(ha_j)$, where $t'$ is a finite iteration number after which $a_i$ is never taken.*

**Proof:** Let us assume that there exists an iteration time $t$ that is greater than $t'$ where $a_i$ is never taken again for simulation. Consider the expression $I\text{-}UCB_t(ha_i) - I\text{-}UCB_t(ha_j)$, which is equal to:

$$I\text{-}UCB_t(ha_i) - I\text{-}UCB_t(ha_j) = \left[\mathcal{V}_t(ha_i) - \mathcal{V}_t(ha_j)\right] +$$

$$(1 - \alpha_t(h))\sqrt{\ln(\mathcal{N}_t(h))}\left[\frac{1}{\sqrt{\mathcal{N}_t(ha_i)}} - \frac{1}{\sqrt{\mathcal{N}_t(ha_j)}}\right] + \alpha_t(h)\left[\hat{\mathcal{H}}_t(ha_j) - \hat{\mathcal{H}}_t(ha_i)\right]$$

Trivially, $\left(\frac{1}{\sqrt{\mathcal{N}_t(ha_i)}} - \frac{1}{\sqrt{\mathcal{N}_t(ha_j)}}\right) \to \frac{1}{\sqrt{\mathcal{N}_{t'}(ha_i)}}$ as $\mathcal{N}_t(ha_j) \to \infty$, since $\frac{1}{\sqrt{\mathcal{N}_t(ha_j)}} \to 0$ and $\mathcal{N}_t(ha_i)$ will be constant under our assumption. Note also that $\frac{1}{\sqrt{\mathcal{N}_t(ha_i)}} - \frac{1}{\sqrt{\mathcal{N}_t(ha_j)}}$ is increasing and will become positive at some time $t^*$ when $\mathcal{N}_{t^*}(ha_j) > \mathcal{N}_{t^*}(ha_i)$. After then (i.e. $t > t^*$), the term $\sqrt{\ln(\mathcal{N}_t(h))}\left(\frac{1}{\sqrt{\mathcal{N}_t(ha_i)}} - \frac{1}{\sqrt{\mathcal{N}_t(ha_j)}}\right)$ will diverge to $\infty$ as $t \to \infty$, $\ln(\mathcal{N}_t(h)) \to \infty$. Further, note that, in a single time step, the maximum difference in rewards is bounded by $r_{max}$, therefore $\mathcal{V}_t(ha_i) - \mathcal{V}_t(ha_j) \ge -\sum_{k=0}^{\infty} \gamma^k r_{max} = \frac{-r_{max}}{1-\gamma}$ (This is because $|\mathcal{V}_t(ha_i) - \mathcal{V}_t(ha_j)| \le \frac{r_{max}}{1-\gamma}$, as shown in Theorem 3 and this implies that $\frac{-r_{max}}{1-\gamma} \le \mathcal{V}_t(ha_i) - \mathcal{V}_t(ha_j) \le \frac{r_{max}}{1-\gamma}$) and $\alpha_t(h)(\hat{\mathcal{H}}_t(ha_j) - \hat{\mathcal{H}}_t(ha_i)) \ge -1$ as $0 \le \alpha_t(h) \le 1$ and $0 \le \hat{\mathcal{H}}_t(h) \le 1$. Therefore, $\exists t''$ where $I\text{-}UCB_{t''}(ha_i) - I\text{-}UCB_{t''}(ha_j) \ge 0$. ■

**Theorem 1** *For any non-empty history $h \ne \emptyset \in \mathbf{H}$ and $\forall a \in \mathbf{A}$, as $\mathcal{N}(h) \to \infty$, $\mathcal{N}(ha) \to \infty$, we have that all states $b$ in $\mathcal{B}(h, b)$ which $P(s = b \mid h) > 0$ will be visited infinitely many times.*

**Proof:** Let us assume that *not* all actions are taken infinitely many times. There must exist *at least* one action $a_i \in \mathbf{A}$ that is taken a finite number of times and, hence, $\exists t'$ after which the action $a_i$ will never be taken. Now, consider any action $a_j$ that is taken infinitely many times as $\mathcal{N}(h) \to \infty$. In Lemma 2, we show that $\exists t'' > t'$ such that:

$$I\text{-}UCB_{t''}(ha_i) > I\text{-}UCB_{t''}(ha_j)$$

We know that the second term in I-UCB grows faster for $a_i$ than $a_j$, as seen from the I-UCB function, since $\frac{1}{\mathcal{N}_t(ha_i)}$ is constant for $t > t'$. Consequently, we reach a conclusion that $a_i$ is chosen over $a_j$ again after $t'$. This further means that for all actions $a_k$'s, we can find a such a finite time instant after which $a_i$ is preferred over $a_k$. Therefore, at some point, the action $a_i$ has to be chosen over all others. This is a contradiction to our original claim. Therefore, all action $a \in \mathbf{A}$ are taken infinitely many times and, hence, all states will be visited infinitely many times. ∎

**Theorem 2** *IB-POMCP's nodes converge to the optimal value for a fixed horizon $D$ as the number of search iterations goes to infinite (i.e., $\mathcal{N}(h_{\mathfrak{r}}) \to \infty$).*

**Proof:** We adapt the proof from the UCT convergence in Shah et al. 2020 and POMCP convergence in Silver and Veness 2010. We induct over the depth of the tree, starting from the leaf up to the root. From Theorem 1, the leaf nodes will have an unbiased estimation of the expected reward $r_h^a = \mathbb{E}_{b \in \mathcal{B}(h)}(\mathcal{R}(b, a)), \ \forall a \in \mathbf{A}$. Note that we are using $r_h^a$ as the expected reward given a $h$, which is different from an immediate reward $r_b^a = R(b, a)$ given a state $b$. Similarly, any state at level $D - 1$ has a simple UCB-like problem since each state is visited infinitely many times and the one-step rewards have an unbiased estimation. In detail, consider the leaves at level $|h| = D$, as $\mathcal{N}(h_{\mathfrak{r}}) \to \infty \Rightarrow \mathcal{N}(h) \to \infty, \alpha \to 0$, and in the absence of children's entropy $\mathcal{H}(ha)$ (as it is the leaf node and has no children), $r_h^a$ converges unbiasedly $\forall \ h \in \mathbf{H}, a \in \mathbf{A}$. Consequently, $\mathcal{V}(h) \to \mathcal{V}^*(h)$ since it is a single decision process, akin to multi-armed bandits. Now, we assume convergence for all levels from the leaves up to $|h'| = |h| + 1$, and as we know $r_h^a = \sum_{s \in \mathbf{S}} P(s_t = b|h) r_b^a$. As we know, the value function satisfies:

$$\mathcal{V}^*(h) = \max_{a \in \mathbf{A}} \left[ r_h^a + \gamma \sum P(h_{t+1} = h'|h_t a_t = ha) \mathcal{V}^*(h') \right]$$

From our hypothesis, the $\mathcal{V}^*(h')$ must converge and moreover, through the sampling process, the value function must converge for level $|h|$ [32] . Consequently, by backward induction, the whole tree's values must converge. ∎

**Theorem 3** *Given any $\epsilon > 0$, there is a finite horizon $D$ for the problem, such that $|\mathcal{V}^*(h_{\mathfrak{r}}) - \mathcal{V}_D^*(h_{\mathfrak{r}})| \leq \epsilon$, where $h_{\mathfrak{r}}$ represents the root of the tree.*

**Proof:** Consider $\pi^*$ and $\pi_D^*$ to be the optimal policies for the infinite and finite horizon problem, respectively. Also, assume that $\epsilon$ *is an arbitrary, positive real number*. Now, let the value functions for these problems be $\mathcal{V}^*(h_{\mathfrak{r}})$ and $\mathcal{V}_D^*(h_{\mathfrak{r}})$. For the root $\mathfrak{r}$, we can find that:

$$|\mathcal{V}^*(h_{\mathfrak{r}}) - \mathcal{V}_D^*(h_{\mathfrak{r}})| \leq \sum_{k=0}^{\infty} \gamma^k r_{max} = \frac{\gamma^D}{1 - \gamma} r_{max} \leq \epsilon$$

since we can calculate $D = \log_\gamma \frac{\epsilon(1-\gamma)}{r_{max}}$, we can also ensure that the relation holds. ∎

**Corollary 1** *IB-POMCP converges to $\epsilon-$optimal actions at the root ($h_{\mathfrak{r}}$) in a $\gamma$-discounted infinite horizon scenario.*

**Proof:** From Theorem 6, we can always find a depth $D$ such that the value function of the finite-horizon problem is arbitrarily close to the infinite-horizon problem. From Theorem 5, we know that we can converge to the optimal value in a finite horizon POMDP. Therefore, since $\alpha \to 0$ as $\mathcal{N}(h_{\mathfrak{r}}) \to \infty$, IB-POMCP will converge to $\epsilon-$optimal actions by choosing $D$ appropriately. ∎

## C   Benchmark Domains and Scenarios

All environments used were obtained from the literature. The Tiger problem is a well-known standard problem [36]. For the Maze domain, we based our design on Thomas et al. 2020 [36]. For the RockSample problem, we designed our own scenarios but based the implementation on Thomas et al. 2020 [36]. For Tag and LaserTag, we used the scenarios proposed by Somani et al. (2013) [34]. For Foraging [2], we proposed our own configurations for each scenario.

**Tiger Environment:** The **Tiger** problem is a classical benchmark where an agent *must choose between two doors: one hides a treasure; the other, a tiger.* The objective is to find out which door hides the treasure. The agent can decide which door to open or wait and listen to the tiger (with a certain probability of mishearing it behind one of the doors) before making the final decision.

The agent has a $15\%$ probability of mishearing the tiger. A maximum of 20 actions are allowed per experiment. Choosing the right door gives the agent a reward of $+0.1$, the wrong door $-1$, and the listening action penalises the agent with a $-0.01$ reward. Thus, each agent could try to listen to the tiger 19 times before making the final decision to open either the left or right door.

**Maze Environment:** This is an *Active Localisation Problem* where an agent navigates through a toroidal grid and *tries to localise its own position* by collecting information about its current position. The reward is based on the entropy of the current belief. There is a change of $15\%$ to miss observe the colour of the cell. The belief follows a Bayesian Update process. Figure 1 illustrates the scenarios.

(a) **Maze Cross –** This is a 5x5 scenario in which black cells are positioned to ease the agent's localisation problem. Figure 1a illustrates the scenario's configuration.

(b) **Maze Holes –** This is an 8x8 scenario, which requires the agent to reason one step in advance to search for the missing black cell in this regular configuration. Figure 1b illustrates the scenario.

(c) **Maze Dots –** This is a 6x6 scenario, identical to Maze Hole except that black cells are separated by several white cells. Figure 1c illustrates the scenario.

(d) **Maze Grid-X –** This is a 3x3 scenario, with an upper white triangle and a bottom black triangle for the localisation. Figure 1c illustrates the scenario.

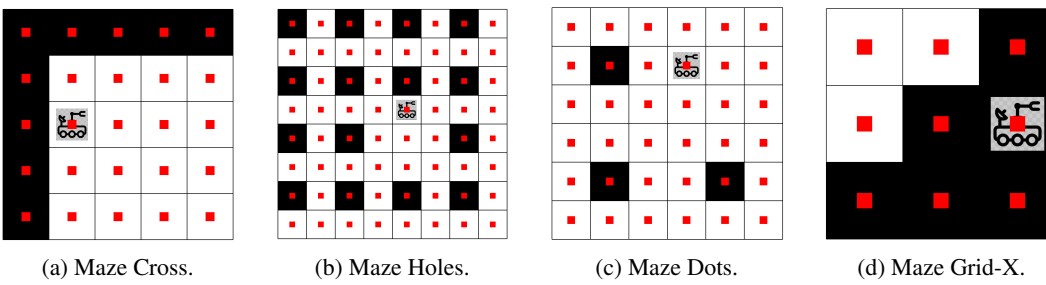

(a) Maze Cross.  (b) Maze Holes.  (c) Maze Dots.  (d) Maze Grid-X.

Figure 1: Maze environment scenario's configuration.

**RockSample Environment:** The context of this problem illustrates a rover exploring an unknown planet. The rover's objective is to earn rewards by both sampling rocks in the environment and leaving the planet with the samples. While the positions of the rover and the rocks are known, not all rocks hold scientific value, which are referred to as "good" rocks. Given the cost associated with rock sampling, the rover is equipped with a noisy long-range sensor, enabling it to assess a rock's potential scientific value before deciding whether to approach and sample it. The environment is a grid map of size $N \times N$ with $k$ rocks. The POMDP model for $RockSample(N, k)$ follows: The state space is a combination of $k + 1$ features, where $Positions$ represent the rover's location $\{(1, 1), (1, 2), ..., (N, N)\}$, and $k$ binary features, $RockType_i$, indicating whether each rock is "Good" or "Bad". There is a terminal state (portal) at the top-right corner of the map. The rover has a choice of $k + 5$ actions: $\{North, South, East, West, Sample, Check_1, ..., Check_k\}$. The first four are deterministic single-step motion actions. The $Sample$ action involves sampling the rock at the rover's current location. If the rock is determined to be "Good", the rover receives a reward of $+1$, and the rock transitions to "Bad", signifying no further benefit from sampling. If the rock is "Bad", a penalty of $-1$ is delivered to the agent. Moving into the exit portal yields a reward of $+0.0001$. All other actions have no associated cost or reward. Each $Check_i$ action employs the rover's long-range sensor to observe $Rock_i$, yielding a noisy observation of either "Good" or "Bad". The noise in the long-range sensor reading is influenced by the efficiency $\eta$, which decreases exponentially as a function of Euclidean distance from the target, following $\eta = exp(-0.2 \, EuclideanDistance(rover, Rock_i))$. Initially, every rock is assumed to have an equal probability of being "Good" or "Bad". Figure 2 shows the four scenario configurations used.

(a) **RockSample22** – This scenario is set on a 5x5 grid, featuring 2 good rocks and 2 bad rocks. The rover initiates its mission from the middle of these rocks and must locate each specific good rock among the bad rocks. Figure 2a illustrates the scenario's configuration.

(b) **RockSample40** – In this 5x5 scenario, there are 4 good rocks initially placed in the environment, with no bad rocks present. In contrast to RockSample22, the rover's goal here is to collect more rewards as there are more good rocks available. The rover starts its journey from the centre of these rocks. Figure 2b illustrates the scenario's configuration.

(c) **RockSample44** – This 10x10 scenario features 4 good rocks and 4 bad rocks. The rover's starting point is the bottom-left corner of the map. Similar to RockSample22 but on a larger scale, the rover's challenge is to locate each specific good rock among the bad rocks to optimize its reward. Figure 2c illustrates the scenario's configuration.

(d) **RockSample17** – In this 10x10 scenario, there's only 1 good rock and 7 bad rocks, and the rover begins its journey in the bottom-left corner of the map. Unlike RockSample44, the expectation here is for the rover to collect fewer rewards. However, if the rover finds the only good rock available, it can significantly improve its performance in terms of reward collection. Figure 2c illustrates the scenario's configuration.

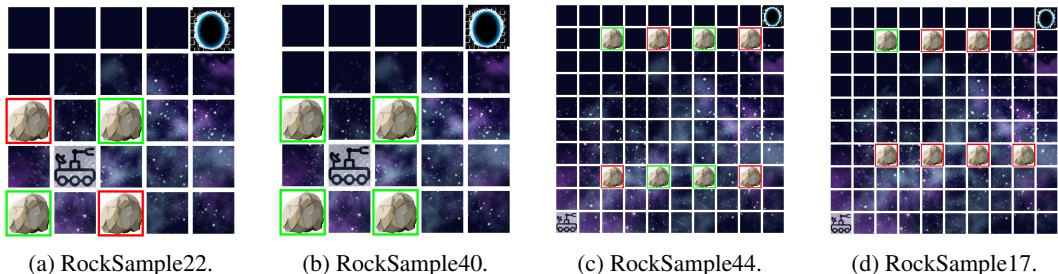

(a) RockSample22.  (b) RockSample40.  (c) RockSample44.  (d) RockSample17.

Figure 2: RockSample environment scenario's configuration.

**Tag and LaserTag Environment:**  In the Tag scenario (Figure 3a), the agent's primary objective is to locate and tag a target that actively moves away. Both the agent and the target navigate within a grid featuring 29 possible positions. While the agent is aware of its own location, it can only observe the target's position when they occupy the same spot. The agent has the option to either stay in its current position or move to any of the four adjacent positions, incurring a cost of $-0.1$ for each move. Additionally, it can opt to execute the tag action, wherein it receives a reward of $+1$ for a successful tag but receives a penalty of $-1$ if it fails to tag the target.

LaserTag (Figure 3b) is an augmented version of Tag, where an agent is acting in a 7x11 grid world containing randomly placed obstacles. The agent's behaviour and that of its opponent align with the rules established in Tag. However, in LaserTag, there's a key distinction: the agent possesses prior knowledge of its initial location. Additionally, the agent is equipped with a laser system capable of providing distance estimations (in cells' units) in 8 directions. The laser readings are generated from a normal distribution centred around the agent's true distance from the nearest obstacle in each direction, with a standard deviation of 2.5 units. These readings are then discretised into whole units, resulting in an observation comprising a set of 8 integers. Figure 3 illustrates the scenarios.

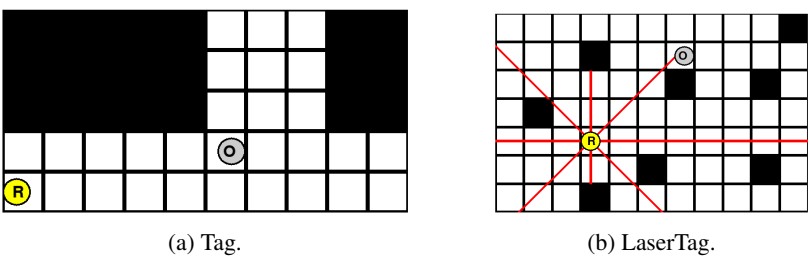

(a) Tag.          (b) LaserTag.

Figure 3: Tag and LaserTag scenarios' configuration.

**Foraging Environment:** This is a common problem used for evaluating online planning algorithms [2, 7]. This domain presents *an agent that must collect boxes displaced in a rectangular grid-world*. The problem is defined over partial observability and the agent does not know how the tasks are distributed. The problem *ends when the agent collects all boxes*. A reward of $+1$ is delivered to the agent every time a box is collected. The radius and the angle of vision (integer numbers) for the agents are, respectively, $20\%$ of the diagonal dimension in the environment in cells' unit and $90^o$ – for example, if we have a $10 \times 10$ environment, so our vision radius will be $radius = 0.2\sqrt{x_{dim}^2 + y_{dim}^2} = 0.2\sqrt{100 + 100} = 0.2\sqrt{200} = 2$. Additionally, agents have memory, i.e., after seeing (in the real world) a state or position in the environment, they will sample the next possible world configuration considering this information. Moreover, obstacles block the agent's vision, hence, it can not see tasks across the walls. Figure 4 illustrates the scenarios.

(a) **The Corridor –** This is a simple $(20, 2)$ scenario, where the agent starts in the left part of a corridor with a box at its side and another at the end of the corridor. The idea here is to test the agent's capability to find isolated reward spots. Figure 4a illustrates the scenario.

(b) **U-Shaped –** A complex version of The Corridor, the U-shaped is a $15 \times 15$ scenario, where the agent starts at the tip of a U-shaped corridor with a box at the beginning, one in the middle and another at the end. The idea here is to test the agent's capability to perform simple planning, but facing sparse rewards collection. Figure 4b illustrates the scenario.

(c) **U-Obstacles –** A complex $20 \times 10$ scenario, where the agent needs to collect the boxes distributed in a room, but there are obstacles (U-shaped walls) that block its vision. This scenario is commonly used to study local minima issues that emerge in robotic navigation problems [22]. The idea here is to test the agent's capability to perform planning with sparse reward collections, and the vision is blocked by obstacles. Figure 4c illustrates the scenario.

(d) **The Warehouse –** This scenario is our largest in terms of dimensions, $20 \times 20$. The agent's objective is to locate specific reward spots (groups of boxes). These reward clusters are situated at considerable distances from one another. Consequently, the agent cannot observe multiple reward clusters from a single position, making planning more challenging. Figure 4d illustrates the scenario.

(e) **The Office –** Our most complex $15 \times 10$ scenario, where the agent needs to collect the boxes distributed in an office. The idea here is to test the agent's capability to perform complex planning when tasks are distributed in different rooms and its vision is blocked by walls. The agent needs to enter each room to finish the problem. Figure 4e illustrates the scenario.

## D   Hyperparameters

We used a single hyperparameter set for all Monte-Carlo Tree Search-based methods:

- Historical weight/Discount factor: $\gamma = 0.95$
- Maximum depth for the tree: $20$
- Maximum number of simulations: $250$ (per search)

For $\rho$-POMCP and TB $\rho$-POMCP specific hyperparameters, we consider the following:

- Small bag size: $|\mathfrak{B}| = 10$
- TB $\rho$-POMCP runs within the same time window of IB-POMCP's average planning time.

Finally, considering IB-POMCP practical enhancement, we applied $q = 0.2$, hence $\alpha \in [0.2, 0.8]$.

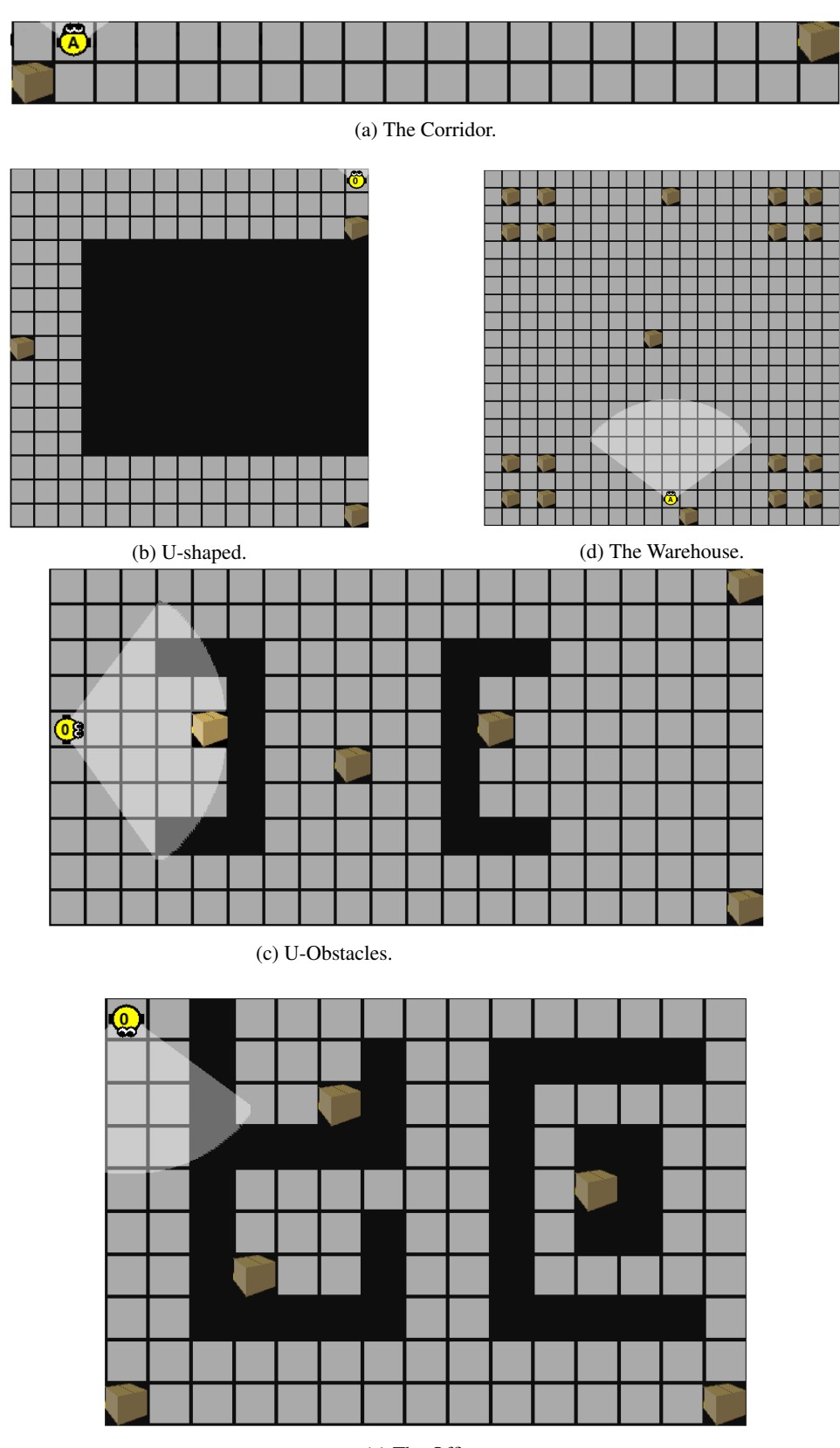

(a) The Corridor.

(b) U-shaped.

(d) The Warehouse.

(c) U-Obstacles.

(e) The Office.

Figure 4: Foraging environment scenario's configuration.

