# OpenReview forum: "Information-guided Planning: An Online Approach for Partially Observable Problems"
_NeurIPS.cc/2023/Conference — NeurIPS 2023 poster_

### Official Review · Reviewer_hPBV · 2023-06-29

**Soundness:** 3 good
**Presentation:** 3 good
**Contribution:** 3 good
**Rating:** 6
**Confidence:** 4

**Summary:**

The paper presents an extension to the POMCP algorithm for online planning under partial observability, called Information-Based POMCP (IB-POMCP). The distinguishing feature of IB-POMCP is that it uses information entropy to enhance the tree search procedure. One effect of this extension is that IB-POMCP is better able to handle sparse rewards, performing its search based on which branches of the tree are likely to yield higher information gain, unlike previous methods which consider only exploration of the tree.

The first modification to POMCP is that when an action is taken in IB-POMCP, new particles are sampled after an action is taken, depending on how unlikely the resulting observation is. If an observation is received that had a high probability, then the particles in the new b root node are expected to capture the belief well. On the other hand, if the probability is low, then more new particles are generated from a uniform distribution. The intention of this is to improve the algorithm’s starting point at each step.

The second modification is the introduction of a new exploration coefficient which is recalculated at each iteration of the search - replacing the alpha coefficient from UCB1 with a new version for I-UCB. The I-UCB alpha coefficient captures information about visit counts similar to UCB1, but includes an additional term which favours nodes with high entropy. The idea of this term is to seek information gain during the search procedure, where the entropy is estimated from the set of observations seen so far. This is a more nuanced search procedure than standard UCB1, which only considers the number of times each node has been visited, but not the distribution of observations received.

**Strengths:**

The paper presents an approach to planning for partially observable problems that makes a clear original contribution to the literature on planning under partial observability.

The authors justify the benefits of their information-guided approach to POMCP-style planning in a manner that makes intuitive sense. The paper also provides a strong experimental justification for the approach, evaluating against relevant baselines in standard domains, as well as providing an ablation study that justifies the individual components of their algorithm. The experiments cover a good range of domains, and the intuition behind the results is explained fairly well.

The paper is overall well structured and written, and presents its key ideas in a clear and logical manner.

**Weaknesses:**

After discussion, I believe that I was a bit too negative regarding the substantiality of the contribution, so I'm increasing the score. Please do not that the section with the theoretical results needs improvement. I think adding intuition and discussion around those results is more important than adding a new domain to the experiments.

---------
My main concern with the paper is the incremental aspect of its contributions. The two improvements over POMCP are (i) adding some standard tricks to the particle filter so that it tracks the belief more accurately, and (ii) a new action selection equation which simply considers entropy of the belief as an extra term. This is a bit on the light side. I’d expect a strong theoretical analysis and justification of Equation (3) to bring the significance of the contribution up. Furthermore, the technical section is simply a listing of results without any context or intuition - not enough insight is given to the reader.

Other issues:

* Equation 2 could do with more justification/explanation of how the update works.
* Line 128 the terminology “the probability of stepping into this node” is unclear and needs further explanation at this point in the text. It made sense to me only after I got up to Section 4(ii).
* Figure 1 caption should explain in more detail what is being shown (e.g. “bright red dots represent X, faded red dots represent Y, blue dots represent Z”). This is mentioned in the text in Line 232 but more detail would be helpful to clarify the diagram.
* Following from the above, I think the authors should make a clearer distinction between the (online) planning phase and action execution. For example, I don’t think it’s clear from the paper that one *only* updates the belief after executing an action, and the POMCP tree keeps histories as nodes but does not do any explicit belief update.
* Lines 264-270 could do with a clearer explanation of the terms “information gain” vs “exploration gain”.

**Questions:**

* Please argue for the significance of the contribution presented in the paper. How is this a substantial step from the state-of-the-art?
* Unclear to me why $\alpha(h_\tau)$ (Equation 1) is defined only at the root node rather than at every subsequent node - is this to make the algorithm simpler, or is there an actual justification for why this is preferable or doesn’t matter?
* Line 351-354 is a bit of a long-winded sentence and unclear what is meant by the final bit - “often seek paths that lead to spots with high entropy in order to decrease uncertainty”, what uncertainty is being decreased here?
* Line 369 what is meant by “without penalising reasoning time”? Is this just saying that IB-POMCP performs well without taking longer to plan than other methods, or something more specific?

**Limitations:**

The experiments section might be strengthened by the addition of another baseline which uses some sort of latent world model (e.g. SARSOP?). It would be interesting to see a comparison with a model that does better than IB-POMCP on performance but requires much more computation time to do so. This would provide a concrete justification for the claim in Lines 51-55 that such modelling is too computationally intensive.

---

> ### Author Rebuttal · Authors · 2023-08-09
>
> We thank your feedback and comments towards acceptance.
>
> ### 1.“Argue for the significance of the contributions.”
> - **Weak constraints to run**: we refined action selection (via I-UCB) and proposed a novel framework (IB-POMCP) that *does not demand prior data collection or the modelling of latent functions*.
> - **Handles sparse rewards**: IB-POMCP is the pioneer in leveraging entropy to estimate information gain using both real-world and internally generated observations, allowing it to identify promising states even *when rewards are not* readily *available* or *sparse* within the planning horizon.
> - **Light-weight framework**: we *enhance planning without increasing reasoning time*. Our novel approach potentially impacts AI and decision-making by improving uncertainty planning capabilities while maintaining efficiency.
>
> ### 2.”My main concern with the paper is its incremental aspect.”
> - IB-POMCP and I-UCB are not incremental; rather, they represent a **well-connected and innovative approach** to enhancing POMCP planning and reasoning.
> - Our approach significantly improves the belief tracking in the particle filter, surpassing standard techniques in the literature by **(i)** using our refined weight coefficient, $\alpha$ ; **(ii)** using our I-UCB equation for tree expansion, and; **(iii)** using our novel strategy to update the root’s particle filter based on tree search result.
> - Our method delivers accurate and efficient planning, *demonstrated empirically and theoretically*.
> - Incorporating entropy-based weight factors into the action selection equation and particle update is a unique and novel contribution. This drives informed decisions and substantial performance improvements, even in scenarios without available rewards within our reasoning horizon. *Each individual contribution is combined in a non-trivial and novel manner*, and their combined effect, as showcased in the Ablation Study results (Section 6), significantly enhances the overall performance.
>
> ### 3.”The authors add some standard tricks to the particle filter to track belief more accurately”
> - Our particle filter update proposal *does not implement any existing method or trick to improve the belief’s track*. Our proposed solution is the result of our information-guided search process, hence the I-UCB implementation, another novel solution in the paper.
> - Refer to answers (1) and (2).
>
> ### 4. “The new action selection equation simply adds the entropy as an extra term.”
> - The addition of our extra term required the *mathematical design and tuning of a **non-trivial** part of the equation*. This process required the study of how observations behave inside the tree search framework and the proposal of an equation that handles the estimation in a way to improve performance.
> - Also, refer to answers (1) and (2).
>
> ### 5.”What is meant by “without penalising reasoning time”?”
> - We mean that IB-POMCP performs well without taking longer to plan than other methods. The reasoning time considers the whole search process until the algorithm delivers the best action to the agent.
>
> ### 6.“I’d expect a strong theoretical analysis of Equation (3)”
> - Our complete theoretical analysis (Technical Appendix) shows IB-POCMP convergence to epsilon-optimal action based on the analysis of Eq 3.
> - Lemma 1 addresses the alpha coefficient convergence in order to show, in Lemma 2, that Eq 3 converges for a finite number of iterations.
> - The consequence of the convergence of Eq 3 is our Corollary 1.
>
> ### 7.”Experiment section might be strengthened by another baseline (e.g. SARSOP?). It would be interesting to see a comparison with a model that does better than IB-POMCP on performance but requires much more computation time to do so.”
> - SARSOP is an off-line planner already compared to POMCP in the original POMCP paper, where it was shown that POMCP achieves similar results with a significantly reduced computational time.
> - $\rho$-POMCP doesn't take as much time as SARSOP, and it is a more recent algorithm, serving to present the suggested comparison. Besides, it demands additional knowledge to run than IB-POMCP (transition and observation function probabilities).
> - $\rho$-POMCP and IB-POMCP share the same base algorithm (POMCP), but our approach stands out in its ability to handle scenarios with sparse rewards. We demonstrate competitive results in terms of reward collection (refer to Table 1).
>
> ### 8.”Why $\alpha(h_r)$ is defined only at the root node?”
> - To clarify, $\alpha$ is defined at the root because we update it at the root’s level. However, its calculated value is used in the action selection of all nodes.
> - This decision was made after exploring various approaches and collecting preliminary results. Updating $\alpha$  solely at the root showed better results than updating it for every node in the simulation path.
> - Our update strategy considers the complete simulation’s information to update $\alpha$ (all rewards and observations collected through the simulation path). Hence, our update occurs after **all this information is backpropagated** to the root.
> - Updating $\alpha$ for all nodes in the simulation path significantly enhances computational efficiency ($100$ updates in a tree with depth $10$ and $100$ simulations, would be $1000$ updates in the worst-case scenario for the all-nodes approach).
>
> ### 9.”“Often seek paths with high entropy in order to decrease uncertainty”, what uncertainty is being decreased?”
> - The system’s entropy is being decreased. In Maze, the agent must seek cells that maximise entropy to maximise the reward collection. As demonstrated in the results, IB-POMCP finds these cells frequently and then collects the higher reward (reducing the entropy).
> - We can make it clearer by refining the text.
>
> ### 10. Minor issues
> - We will fix all the minor points mentioned in your review: figure caption, term clarification, etc.
> - Regarding Equation 2, we will add further detail in our supplementary material.

---

> > ### Comment · Reviewer_hPBV · 2023-08-14
> >
> > Thank you for the response.
> >
> > I liked the three-point summary of the relevance of the approach, thanks. T
> >
> > It seems to me that you do particle reinvigoration when one moves to a node $ haz$ in the tree that wasn't sampled often from $ha$. How is that related to $\alpha$ or I-UCB, as you claim in your response? Those terms are used for action selection, but they don't seem to be used for sampling.
> >
> >
> > Regarding the formal analysis, I don't think a section that simply states the results but doesn't discuss them at all or gives intuition on how they are used is suitable.

---

> > > ### Author Response · Authors · 2023-08-16
> > >
> > > Dear hPBV reviewer, thank you for your reply.
> > >
> > > ### Particle reinvigoration Novelty
> > >
> > > Our Root Update step can indeed be interpreted as a particle reinvigoration process, yet it is important to note that  **our information-guided planning affects this process**.
> > >
> > > The Root Update process is intricately connected to the outcomes of our adaptive search strategy, particularly in relation to the parameter $\alpha$ and its embodiment into the I-UCB equation. I-UCB can effectively guide the search process towards promising states using the $\alpha$ value, which adapts the search and simulation policy within the tree. Our approximation of the weight used in the Root Update $\tilde{P}(z|h_r a)$ considers the node visit counts (Line 169), which is an inherent reflection of the agent's interest in expanding particular branches.
> > >
> > > In simpler terms, our Root Update process is intentionally designed to leverage the information gain generated from both simulated world scenarios in the agent's mind and real-world observations. This synergistic approach serves to enhance the entire online planning process. In practice, this involves considering the executed action and real-world observation to navigate within the tree and the outcomes of our information-guided search to invigorate our belief in the particle filter.
> > >
> > > To offer an analogy, much like how humans tend to explore promising avenues when pondering a problem, our computational approach mirrors this by predominantly focusing on potential high-value states, in terms of rewards and information gain. Subsequent to taking an action, we evaluate whether the result aligns with our anticipated outcomes, and quantify the likelihood of achieving that result. Our computational Root Update operation approximates this probability by considering the frequency with which particular outcomes were encountered during the search process (observations) and then defines confidence in its planning process (if I frequently explore a state and often found the same outcome, I might be correctly approximating the world that I’m acting on and collecting knowledge). This confidence is the key to our weighting process and updating the root’s particle filter.
> > >
> > > Following this analogy, it is possible to see that both solutions are connected in a non-trivial and novel manner. An adaptive search policy is indispensable in realizing the benefits of our update process in the particle filter. As indicated by our Ablation study, the application of information-guided particle reinvigoration (IPR-POMCP baseline) with the calculation of $\tilde{P}(z|h_r a)$ alone, while incorporating the traditional exploration coefficient $c$, leads to a smaller advancement and lesser performance gains compared to IB-POMCP, which connects the Root Update with the I-UCB equation.
> > >
> > > We will be glad to improve the discussion to make this intuition clearer in the text for the final version of the paper if we are accepted for publication.
> > >
> > > ### Formal analysis
> > >
> > > We are more than happy to improve and extend the discussion of our formal analysis (theoretical results) in the main paper, outlining the implications of these results. We will also add further information in the Technical Appendix.

---

> > > > ### Comment · Reviewer_hPBV · 2023-08-17
> > > >
> > > > I'm still not convinced about the argument about particle reinvigoration and information-guided planning. Information-guided planning allows you to explore better and get better results. I'm fine with that. However, does it allow you to track the belief more accurately? I don't see how. If you had another algorithm doing particle reinvigoration but using another exploration policy (say IPR-POMCP), your analogy would still work, it would just be that I wasn't as smart in the way I chose the actions and states to focus on. However, I'd still focus more on certain actions from the root node, so would have better (simulation-based) estimates of the outcomes of that action. If I then take it and end up observing something I wasn't expecting, then I need to resample particles.
> > > >
> > > >  IB-POMCP might be doing better than IPR-POMCP just because it explores better, not because it does "better" particle reinvigoration. I'd like to see some experiment that clearly shows that by using IB-POMCP you can track the belief better than using IPR-POMCP. If this is not the case, then I think you're exaggerating the claims regarding the synergy between these two things.

---

> > > > > ### Author Response · Authors · 2023-08-18
> > > > >
> > > > > Thank you very much for your message and the opportunity for discussion.
> > > > >
> > > > > Our method employs a dynamic threshold derived from the dynamic value of $\tilde{P}(z|h_r a)$ (or specifically, $1 - \tilde{P}(z|h_r a)$ that determines the ratio of uniform samples in the reinvigorated particle filter). Unlike some traditional approaches which use a fixed value for that threshold, or some other state-of-the-art contributions, which mostly rely on additional knowledge and expensive calculations [1,2,3], IB-POMCP does not need an explicit true distribution of observation probabilities, transition function or training data in this process by using our proposed $\tilde{P}(z|h_r a)$ probability.
> > > > >
> > > > > It's important to note that the action selection process within the planning phase has a direct impact on the $\tilde{P}(z|h_r a)$ value, as previously explained. Therefore, the strategy used to choose actions for simulation during planning will affect the particle filter, since it affects the dynamic threshold $\tilde{P}(z|h_r a)$. Hence, **we can say that the I-UCB equation has an impact on belief tracking**.
> > > > >
> > > > > IPR-POMCP, which also employs our dynamic threshold, shows through our ablation study significant improvement to the conventional POMCP approach, which follows a standard particle reinvigoration paradigm (with a fixed parameter for reinvigoration). Hence, just using the **dynamic threshold** $\tilde{P}(z|h_r a)$ **improves results**. We highlight here that the IPR strategy also represents one of our contributions in this paper.
> > > > >
> > > > > Furthermore, the ablation study indicates that combining the dynamic threshold with the I-UCB equation yields even more favourable results and that using solely the I-UCB equation without the dynamic threshold (IUCB-POMCP baseline) results in worse outcomes than the full IB-POMCP approach. Hence, **both the dynamic threshold and the I-UCB equation are important to improve results**, and their simultaneous application produces even greater enhancements than when employed individually.
> > > > >
> > > > > What is perhaps still left open is whether the I-UCB equation leads to better $\tilde{P}(z|h_r a)$ values than the UCB equation. Since our approach takes the estimated entropy into consideration for action selection we believe that I-UCB may lead to better threshold values.
> > > > >
> > > > > For example, in cases where a node possesses high entropy and no reward in the reasoning horizon, the I-UCB equation could prioritize visiting it more often than the UCB equation would, increasing the $\tilde{P}(z|h_r a)$ value for that particular node (note that it will not imply in worse probability estimation because the entropy is based on the estimation of observations and the particle filter gives the probabilities of states). The high entropy indicates that the node accommodates a more diverse set of particles in the particle filter and hence sampling particles from a uniform distribution during particle reinvigoration (to increase diversity) becomes less useful. Hence, the implications of the I-UCB equation appear favourable as it leads to fewer particle samples from the uniform distribution in this case (as we use the ratio 1 - $\tilde{P}(z|h_r a)$).On the other hand, if a node has low entropy, the particle filter may easily get biased to a small set of states. In this case, I-UCB would lead to a higher number of particles sampled from the uniform distribution than the UCB equation in the particle reinvigoration, since it would have a lower $\tilde{P}(z|h_r a)$ value in I-UCB. Therefore, we believe that I-UCB should be better than UCB for defining the dynamic threshold $\tilde{P}(z|h_r a)$, and hence better in belief tracking.
> > > > >
> > > > > We agree that showing experimentally the potential of I-UCB to improve belief tracking, improving even further our idea of a dynamic threshold, would be a very nice addition to our paper. However, it's important to note that these experiments would require knowledge of the true observation probabilities in an explicit form, in order to compare the alternative particle filters against a given ground truth. We are currently working on these experiments, which will be added to the final version of our paper.
> > > > >
> > > > > [1] Katt, S. et. al. “Bayesian Reinforcement Learning in Factored POMDPs”. In AAMAS 2019
> > > > >
> > > > > [2] Hayashi, A. et al. "Reasoning about uncertain parameters and agent behaviours through encoded experiences and belief planning." In Elsevier Artificial Intelligence 2020
> > > > >
> > > > > [3] Fox D. “Adapting the Sample Size in Particle Filters Through KLD-Sampling “. In IJRR 2003

---

> > > > > > ### Comment · Reviewer_hPBV · 2023-08-21
> > > > > >
> > > > > > Thank you for the clarifications.
> > > > > >
> > > > > > > What is perhaps still left open is whether the I-UCB equation leads to better
> > > > > > > values than the UCB equation. Since our approach takes the estimated entropy into consideration for action selection we believe that I-> > UCB may lead to better threshold values.
> > > > > >
> > > > > > This was my main point, which I thought you were not being very clear about. I appreciate the response and I believe some of the discussions we had here would fit well in the paper, to highlight the contribution better. I also understand that the experiments I asked for are not easy to produce.
> > > > > >
> > > > > > I do think some of your responses had a few statements that were a bit grander than they had to be, please be careful not to oversell things either.
> > > > > >
> > > > > > I'm increasing my score.

---

> > > > > > > ### Author Response · Authors · 2023-08-21
> > > > > > >
> > > > > > > Thank you very much. Indeed it was a very interesting discussion. We will be happy to discuss these points in the paper, to better highlight our contribution.

---

### Official Review · Reviewer_JP1G · 2023-07-06

**Soundness:** 3 good
**Presentation:** 3 good
**Contribution:** 3 good
**Rating:** 6
**Confidence:** 1

**Summary:**

This paper presents IB-POMCP, a novel algorithm for online planning under partial observability, that uses entropy estimation to guide the tree search. The main concept is to replace the traditional explore-exploit balancer (UCB1) with (an IUCB score) which is a weighted mixture of value, exploration score, and entropy score. This idea is motivated by the failure modes of other algorithms tackling POMDP problems: poor performance in the case of sparse rewards and especially sparse rewards that are often delivered outside of the algorithm's planning horizon. IB-POMCP is compared against POMCP and TB rho-POMCP on 3 tasks: Tiger, Maze, and Foraging. The experiments show a clear advantage of IB-POMCP. There is also an ablation study which gives even more insight into the method.

**Strengths:**

The experiments are carefully made (with confidence intervals) and clearly show the advantage of the method over baselines. Also, there is a time comparison, which shows another desired property of IB-POMCP: it is usually faster than baselines. There is an exhaustive discussion of the results. The paper contains an ablation study, which compares the effect of two different aspects of IB-POMCP:
a) information guided particle reinvigoration
b) U-UCB score
It is demonstrated that both of the above contribute to the effectiveness of IB-POMCP and there is an interesting comment about these results.
The paper is well-written, images are helpful. There are mathematical theorems guaranteeing theoretical convergence to optimal solution or near optimal (with a limited planning horizon). Mathematical formulas are explained via insightful comments. The main ideas of the paper is clear simple and easy to uderstand.

**Weaknesses:**

It is impossible for me to describe the scientifically important weaknesses of this paper since it is far outside my area of expertise. I can only provide minor weaknesses:

1. The paper assumes a reader has a lot of prior knowledge in the area of POMDP. It is a problem or not depending on the desired reader. As a person who is not familiar with the topic, I like to have more preliminary comments.
2. Notation may be not entirely clear for a person not familiar with the topic, i.e.  line 169, what exactly means: "haz"?


**Questions:**

1. The tiger environment has a limit of 20 actions. What exactly happens when the agent chooses to listen 19 times in a row? In the 20th step, the action space is different and consists of only choosing the door to open or can the agent listen again?
2. In practice: can the entropy of the whole tree behave non-monotonically over time? If yes, how often it happens?
3. line 159: how exactly is the tree re-initialized?

**Limitations:**

There is little about the limitations of the method. I would like to know for example: can IB-POMCP be used on any POMDP? How the resources needed to solve a task with IB-POMCP depend on the problem's complexity? What is the worst-case scenario performance? What is the memory-usage?

---

> ### Author Rebuttal · Authors · 2023-08-09
>
> We appreciate your insightful feedback and acceptance of our proposal. Your thorough review demonstrates the depth of your evaluation and the meaningful consideration.
>
> ### 1.”The paper assumes a reader has a lot of prior knowledge in the area of POMDP.” and “Notation may be not entirely clear for a person not familiar with the topic.”
> - In the revised version, we will *carefully review the notation* to make it more **accessible and understandable** (Sections 3-4), particularly for those new to the topic.
>
> ### 2.Line 169: what is $h_{r} az$?
> - In the current text, we say “$a$ is the action just taken by the agent in the real world, $z$ the real observation received from the environment, $N(h_raz)$ the number of visits of $h_raz$ node, the new root”. We will add a clarification presenting “the number of visits of the node with history $h_raz$” instead of “the number of visits of $h_raz$ node”.
>
> ### 3.”The tiger environment has a limit of 20 actions. What exactly happens when the agent chooses to listen 19 times in a row? In the 20th step, the action space is different and consists of only choosing the door to open or can the agent listen again?”
> - When the agent chooses to listen 19 times in a row, it receives a reward of $-19$ (since each listen delivers a $-1$ reward). And after the 20th step, the agent can listen again. The action space is still the same.
>
> ### 4.”In practice: can the entropy of the whole tree behave non-monotonically over time? If yes, how often it happens?
> - In fact, *the tree entropy behaves in a non-monotonical way*. However, noticing that it may cause problems in the estimation, **we have designed our approach to effectively handle this situation**. The proposed alpha equation (Equation 1) ensures that the entropy values monotonically decrease as we approach infinity, which *aids in the convergence of our entropy and updates the tree knowledge*. This mechanism is akin to the entropy update within the IB-POMCP tree and the utilization of UCB1/I-UCB equations. By incorporating this method, we can effectively manage the non-monotonic behaviour of the tree entropy.
> - Regarding the frequency of changes in the entropy trend, it depends on the size of our observation space and how often we can visit different states within our reasoning horizon, hence, depends on the internal policies of the tree.
>
> ### 5.”line 159: how exactly is the tree re-initialized?”
> - The re-initialisation of the tree occurs with the creation of a new root node for the tree. We create a new node $(h, V(h), N (h), B(h))$ with $h$ as the current history (sequence of actions and observations collected from the world) $V(h) = 0$, $N(h) = 0$ and $B(h)$ is empty. This new node will be considered as the new root. It means that all other simulations already made in the agent’s head are discarded and we re-start the process using the created node. We refer the reader to [1] for further detail (reference also in the paper).
> - We can add the above explanation in the text to better clarify the tree re-initialisation process.
>
> ### 6.”There is little about the limitations of the method. can IB-POMCP be used on any POMDP?
> - Yes, IB-POMCP can be used on any POMDP. Additionally, we provide theoretical guarantees to its application  (Section 5).
> - Besides that, we can say that all we need is a black-box simulator to sample transitions given a state and an action, since we are based on POMCP, so even an explicit model of the POMDP is not needed.
>
> ### 7.“How the resources needed to solve a task with IB-POMCP depend on the problem's complexity?” and “What is the memory-usage?”
> - IB-POMCP's planning complexity is comparable to POMCP.
> - In terms of time, in most studied cases (see Result section, line 357, for insights on this analysis), we could perform similarly or faster than POMCP.
> - In terms of memory, IB-POMCP will require more memory to allocate the observation multi-set. But, it is still small and it does not impact reasoning time.
>
> ### 8. What is the worst-case scenario performance?
> - We believe that our worst-case performance is comparable to POMCP, hence, UCT, being  $\Omega(exp(exp(. . . exp(1) . . .)))$ [1].
>
> [1] Silver, D. et. al. "Monte-Carlo planning in large POMDPs." In NeuRIPS, 2010.

---

> > ### Comment · Reviewer_JP1G · 2023-08-18
> > **Thanks**
> >
> > Many thanks for the answers. Thanks for all the clarification and for considering the revision of the notation to make it more accessible for people new to the topic.

---

### Official Review · Reviewer_zmgm · 2023-07-06

**Soundness:** 3 good
**Presentation:** 2 fair
**Contribution:** 3 good
**Rating:** 6
**Confidence:** 2

**Summary:**

This paper proposes a novel approach for online planning under uncertainty. It leverages the information entropy to guide a tree search process.

**Strengths:**

* This paper is welled structured.
* The paper conducts both empirical and theoretical analysis.

**Weaknesses:**

N/A

**Questions:**

N/A

---

> ### Author Rebuttal · Authors · 2023-08-09
>
> Thank you for your assessment and for considering our paper for acceptance.
> We are delighted and encouraged by your positive evaluation and enthusiasm for the work we propose.

---

### Official Review · Reviewer_QZzo · 2023-07-07

**Soundness:** 4 excellent
**Presentation:** 3 good
**Contribution:** 1 poor
**Rating:** 3
**Confidence:** 5

**Summary:**

The paper proposes several improvements to the POMCP algorithm. The improvements introduce adaptive sample size, entropy regularized tree exploration and action selection.

**Strengths:**

The research considers an important problem of online planning in POMDP --- dealing with sparse rewards. A few improvements to POMCP are proposed, theoretically analysed and empirically evaluated.

**Weaknesses:**

1) The authors' view on MCTS planning is partial and may be misleading. The authors argue that MCTS is based on UCB1. It is not, UCB1 is just option, there are others. One example is Thompson sampling, https://dl.acm.org/doi/10.5555/3038794.3038799 . Also, AlphaZero does not use UCB1, just as an illustration.

2) The idea of adapting particle set size based on the entropy / KL divergence is nice and useful, but not novel: https://journals.sagepub.com/doi/10.1177/0278364903022012001 . Particle set size should be adapted based on diversity, either intrinsically or extrinsically estimated.

3) The algorithm is not compared with DESPOT, https://papers.nips.cc/paper_files/paper/2013/hash/c2aee86157b4a40b78132f1e71a9e6f1-Abstract.html, the much better performing POMCP variant. There are follow up research publications on DESPOT in the last 10 years, too. DESPOT exhibits much better performance.

4) The domains on which the algorithm is evaluated are very small and are NOT the domains on which POMCP, DESPOT and variants are evaluated for performance in the literature (Tag, LaserTag, RockSample just to name a few examples). I  am not convinced that estimates and modifications in the proposed POMCP variant scale well beyond the toy domains included in the evaluation.

**Questions:**

Belief state particle set reinvigoration involves estimating of the observation given the action.  In all domains used for evaluation, the probability of an observation given the state is given, so this probability can be estimated from the particle state itself rather than from action-observation statistics. How accurate is your estimate compared to the particle set estimate? How does plugging in particle set based estimate affect performance of the algorithm?

**Limitations:**

The authors do not discuss limitations of their proposed algorithm. I suspect that the advantage of the algorithm is limited to toy domains only.

---

> ### Author Rebuttal · Authors · 2023-08-09
>
> Thank you for your insightful feedback and detailed review.
>
> ### 1.“Authors argue that MCTS is based on UCB1. UCB1 is just an option”
> - To avoid confusion, we will clarify that UCB1 is an option by writing that it is "widely used" and adding some variations in the Related Works, such as PUCT (AlphaZero) [1] and the Thompson Sampling-based approach mentioned in this review.
> - Although there are variations, **UCB1 is still a very important algorithm** in the current state-of-the-art for MCTS and ongoing research [2,3,4].
> - E.g., PUCT [1] was inspired by UCB1. It slightly modifies the traditional equation, removing the log function and adding a neural network bias multiplication. This idea modifies the original proposal but still follows the same idea of maximising the upper bound at each node.
>
> ### 2.“Adapting particle set size using entropy/KL divergence is not novel”
> - IB-POMCP does not adapt the particle set size, it modifies the sampling strategy according to an approximation of our uncertainty $\tilde{P}(z|h_r a)$ when updating the root, as presented in Section 4 (i) Root Initialization and Update.
> - The manner in which IB-POMCP incorporates these principles into the POMCP framework is novel and different from KLD strategies.
> - Unlike KLD, IB-POMCP does not need reliable true distribution or training data for optimisation. **Our information-based search eliminates this requirement** by using the search result to approximate our variable which weights the particle filter’s update.
> - Calculating the KL distance, hence the likelihood between distributions requires more computing power than our proposal. **We improve performance without sacrificing reasoning time** (supported by our results). Our sampling strategy uses the above approximation to do so.
>
> ### 3."The algorithm is not compared with DESPOT"
> - POMCP is still a relevant reference in the literature and supports new solutions, as we do [6,7,8,9].
> - We focus on improving POMCP, however, **our contributions can also benefit other algorithms**, including DESPOT-based solutions.
> - We do not compare our results directly with DESPOT, but we do compare our approach against competitive and more recent algorithms: rho-POMCP and TB-rho POMCP [5], both have demonstrated promising results and share the same base algorithm with IB-POMCP (POMCP).
> - DESPOT’s main focus is on filtering a large observation space to plan using a sparse approximation of the problem (smaller). However, **the algorithm still struggles** in large action spaces, and/or **sparse reward scenarios** where the optimal policy could be long (require many steps). Our proposal addresses POMDPs featuring sparse rewards in an efficient manner and hence is focused on solving a different research problem than DESPOT.
> - DESPOT-based algorithms still require the definition of good default policies to run the solution. They also need $D$ and $K$ parameters to be set beforehand, and it would be problematic in the sparse reward case if no or almost no rewards are seen within the $D$ parameter (depth). **Our algorithm, however, can handle this kind of situation**, running the search process from scratch and still leading the tree to expand towards promising high entropy subtrees, even when no rewards are observed in the planning horizon.
> - DESPOT will be added to our related work discussion.
>
> ### 4."The domains are very small and are NOT the domains evaluated in the literature." and “I suspect that the advantage is limited to toy domains only.”
> - Our domains also present relevance in the literature and have been frequently used as benchmarks [9, 10, 11], presenting their own complexity and specific challenges.
> - Tag/LaserTag and RockSample **do not feature the sparse rewards** in their proposal. Instead, rewards are delivered from any state upon performing the "reward collection" action (as in the Maze).
> - For the study of sparse reward, we chose the Foraging environment (details in our Tech. Appendix, Section 3).
> - We also emphasise that our domains complexity is not comparable to simple toy problems. In fact, they are similar to the LaserTag and some of our scenarios are even larger/more complex than this traditional benchmark.
>     - Our Foraging Warehouse scenario, for example, which has dimensions $20\times20$ and $|Z|$ in the order of $10^19$, showcases the scalability of IB-POMCP while surpassing the traditional LaserTag ($7\times11$ with $|Z|$ in the order of $10^6$) as presented in the referred DESPOT's paper.
>
> ### 5."authors do not discuss limitations"
> - We carefully presented the limitations alongside the discussion of our results.
> - We separate the analysis by domain and provided insights into why we performed better and when we might encounter challenges.
>
> [1] Silver, D. et. al. Mastering the game of Go without human knowledge. In Nature 2017
>
> [2] Sunberg, Z. et. al. "Online algorithms for POMDPs with continuous state, action, and observation spaces." In  ICAPS 2018
>
> [3] Hoerger M. et. al. "An On-Line POMDP Solver for Continuous Observation Spaces". In ICRA 2021
>
> [4] Riccio, F. et. al.. "LoOP: Iterative learning for optimistic planning on robots." In RAS 2021
>
> [5] V. Thomas et. al. Monte Carlo information-oriented planning. In ECAI 2020
>
> [6] Amini, S. et. al. POMCP-based decentralized spatial task allocation algorithms for partially observable environments. In Appl Intell 2023
>
> [7] Mazzi, G. et. al. Learning Logic Specifications for Soft Policy Guidance in POMCP. In AAMAS  2023
>
> [8] Zuccotto, M. et. al. Learning state-variable relationships for improving POMCP performance. SIGAPP 2022
>
> [9] Mern, J. et. al. Bayesian Optimized Monte Carlo Planning. In AAAI 2021
>
> [10] Yourdshahi, E et al. "On-line estimators for ad-hoc task execution: learning types and parameters of teammates for effective teamwork." In AAMAS 2022
>
> [11] Schwartz, J. et .al. "Bayes-Adaptive Monte-Carlo Planning for Type-Based Reasoning in Large Partially Observable, Multi-Agent Environments." In AAMAS 2023

---

> > ### Comment · Reviewer_QZzo · 2023-08-12
> >
> > Thank you for your response. Tag/LaserTag definitely feature sparse rewards. In any case, I think evaluating the algorithm on the benchmarks on which the baseline (POMCP) was evaluated is a requirement for acceptance, even if the algorithm demonstrates inferior performance (with appropriate justification).

---

> > > ### Author Response · Authors · 2023-08-14
> > >
> > > Dear QZzo reviewer.
> > >
> > > ### RockSample requirement
> > >
> > > Given your requirement for approval, we revisited the POMCP paper and double-checked the performed experiments in the original POMCP paper [1]. They evaluate POMCP in the RockSample problem. Tag and LaserTag domains are not proposed for evaluation.
> > >
> > > For the submitted paper, we choose Tiger, Maze and Foraging due to their own complexities and challenges presented. Judging them sufficient to show our advancements and considering space constraints (number of pages), we did not include RockSample in our original benchmarks. However, we would be happy to include in the final version, if accepted for publication.
> > >
> > > We tested POMCP, IB-POMCP and TB $\rho$-POMCP in 4 different RockSample scenarios [2]. The results for reward collection are shown in the table below:
> > >
> > >
> > > | Dimension | # of Rocks | POMCP | IB-POMCP | TB $\rho$-POMCP |
> > > | :------------: | :-------------: | :---------: | :-------------: | :------------------------: |
> > > | $ 5 \times  5$ | $2G2B$ | $ 0.006 \pm 0.001 $  | $ 0.007 \pm 0.001 $ |  $ 0.003 \pm 0.001 $  |
> > > | $ 5 \times  5$ | $4G0B$  | $ 0.012 \pm 0.001 $  | $ 0.016 \pm 0.001 $ |  $ 0.011 \pm 0.002 $  |
> > > | $10 \times 10$ | $4G4B$ | $ 0.007 \pm 0.001 $  | $ 0.009 \pm 0.002 $ |  $ 0.006 \pm 0.001 $  |
> > > | $10 \times 10$ | $1G7B$ | $ -0.004 \pm 0.002 $ | $ 0.000 \pm 0.001 $ |  $ -0.006 \pm 0.002 $ |
> > >
> > > The results for reasoning time (in seconds) are shown in the table below:
> > >
> > > | Dimension | # of Rocks | POMCP | IB-POMCP | TB $\rho$-POMCP |
> > > | :------------: | :-------------: | :---------: | :-------------: | :------------------------: |
> > > | $ 5 \times  5$  | $2G2B$ | $ 2.076 \pm 0.035 $ | $ 2.333 \pm 0.037 $ |  $ 5.000 \pm 0.000 $  |
> > > | $ 5 \times  5$  | $4G0B$ | $ 2.084 \pm 0.033 $ | $ 2.399 \pm 0.044 $ |  $ 5.000 \pm 0.000 $  |
> > > | $10 \times 10$ | $4G4B$ | $ 4.671 \pm 0.091 $ | $ 5.104 \pm 0.057 $ |  $ 5.000 \pm 0.000 $  |
> > > | $10 \times 10$ | $1G7B$ | $ 4.721 \pm 0.089 $ | $ 5.107 \pm 0.051 $ |  $ 5.000 \pm 0.000 $  |
> > >
> > >
> > > The scenarios have different 2D dimensions ($D_x \times D_y$ in the first column) and different numbers of good and bad rocks (#G, number of good rocks, and #B, number of bad rocks, in the second column). All algorithms run with the same number of simulations, except for TB rho-POMCP, which reaches its time-out of 5s. (time-budget).
> > >
> > > As shown, IB-POMCP handles well the RockSample scenario, presenting similar advantages as in our studied domains. We show significant improvement in terms of reward collection (p-value < 0.02) in 3 out of 4 scenarios -- p-value < 0.11 in the simplest scenario (RockSample $ 5 \times  5$, $2G2B$). In terms of reasoning time, POMCP is slightly faster than IB-POMCP in 3 out of 4 scenarios (p-value < 0.01).
> > >
> > > If the paper is accepted, we will add these results to our Technical Appendix and discuss further advantages and limitations. We can also present it instead of the Tiger domain in the main paper.
> > >
> > > ### Benchmark selection
> > >
> > > We would like to argue that the replication of all environments used by POMCP back in 2010 should not be an obligatory criterion for publication. The inclusion of more recent works would enhance relevance. We selected some environments based on the insights presented in [3] (ECAI 2020) and we added the Foraging to increase complexity.
> > >
> > > There are instances where advancements have been made upon POMCP and published in respected venues, yet without a comprehensive evaluation across all its environments. For example, [4] studies RockSample and Battery, [5] and [6] introduce new problems (e.g. Lunar-lander and Wind farm) and don't show any experiments in the POMCP’s environments list. Finally, [3] presents only the RockSample as a similar environment to the POMCP paper.
> > >
> > > Some relevant works in AI also do not use the same environments as their classical reference. DQN [7], which notably improved the Q-Learning techniques, shows experiments only in the more recent Atari benchmark, avoiding the same classical environments (e.g., backgammon from TD-gammon [8]).
> > >
> > > That being said, we would be happy to include the results in the RockSample domain in our evaluation, as mentioned above.
> > >
> > > ### Citations
> > >
> > > [1] Silver, et. al. "Monte-Carlo planning in large POMDPs." NeurIPS 2010.
> > >
> > > [2] Alves, M. et. al. Adleap-mas: An open-source multi-agent simulator for ad-hoc reasoning. AAMAS’22.
> > >
> > > [3] V. Thomas et. al. Monte Carlo information-oriented planning. ECAI 2020
> > >
> > > [4] Mazzi, G. et. al. Learning Logic Specifications for Soft Policy Guidance in POMCP. AAMAS 2023
> > >
> > > [5]  Amini, S. et. al. POMCP-based decentralized spatial task allocation algorithms for partially observable environments. Appl Intell 2023
> > >
> > > [6] Mern, J. et. al. "Bayesian optimized monte carlo planning." AAAI 2021.
> > >
> > > [7] Mnih, V. et al. Human-level control through deep reinforcement learning. Nature (2015)
> > >
> > > [8] Tesauro, G. Temporal difference learning and td-gammon. Communications of the ACM, 1995

---

> > > > ### Comment · Reviewer_QZzo · 2023-08-15
> > > >
> > > > POMCP seems to be always faster, if I am looking at the right tables. What would be the results on Tag and/or LaserTag?

---

> > > > > ### Author Response · Authors · 2023-08-18
> > > > >
> > > > > Thank you so much for your message.
> > > > >
> > > > > Yes, POMCP is **slightly** faster than IB-POMCP (about $0.3$ seconds) in RockSample.
> > > > >
> > > > > Considering your curiosity to see the results of IB-POMCP in Tag and LaserTag, we ran the experiments in both and the results follow below.
> > > > >
> > > > > - Reward
> > > > >
> > > > > | Scenario                    | POMCP                     | IB-POMCP               | TB $\rho$-POMCP     |
> > > > > | :--------------------------: | :--------------------------: | :-------------------------: | :---------------------------: |
> > > > > | Tag                            | $ -0.047 \pm 0.009 $ | $ -0.036 \pm 0.007 $ | $ -0.065 \pm 0.009 $ |
> > > > > | LaserTag                   | $ -0.090 \pm 0.006 $ | $ -0.092 \pm 0.006 $ | $ -0.086 \pm 0.009 $ |
> > > > >
> > > > > - Time (in seconds)
> > > > >
> > > > > | Scenario                    | POMCP                   | IB-POMCP               | TB $\rho$-POMCP    |
> > > > > | :--------------------------: | :------------------------: | :-------------------------: | :--------------------------: |
> > > > > | Tag                            | $ 1.220 \pm 0.201 $ | $ 0.862 \pm 0.138 $ | $ 3.000 \pm 0.000 $ |
> > > > > | LaserTag                   | $ 1.763 \pm 0.119 $ | $ 1.482 \pm 0.099 $ | $ 3.000 \pm 0.000 $ |
> > > > >
> > > > > For the Tag scenario, IB-POMCP presents significant improvement in terms of reward collection against both baselines (p-value < 0.04). In terms of reasoning time, IB-POMCP presents a slightly faster reasoning time than POMCP but it is significant (p-value < 0.03). It is due to a similar feature presented in line 357: IB-POMCP simulates transitions in rollouts (#rollouts) more often than transitions within the tree (#transitions) in comparison to POMCP (in a ratio of #rollouts/#transitions = 0.61 for POMCP and 1.7 for IB-POMCP).
> > > > >
> > > > > For the LaserTag scenario, we have a tie between all the methods, with no statistically significant difference spotted in terms of reward collection (p-value < 0.81 for IB-POMCP vs POMCP and p-value < 0.32 for IB-POMCP vs TB $\rho$-POMCP). In terms of time, we also do not have any significant difference between IB-POMCP and POMCP (p-value < 0.56). IB-POMCP is significantly faster than TB $\rho$-POMCP, which reaches its 3s time limit (p-value < 0.06).
> > > > >
> > > > > If the paper is accepted, we will add these results to our main paper and discuss further advantages and limitations.

---

### Author Rebuttal · Authors · 2023-08-10

We thank all the reviewers for their valuable feedback. In this global response, we summarise all the major points  of our work based on this rebuttal phase’s feedback.

### Novelty
- **Particle filter update**
   - Our root's particle filter update strategy is based on the behaviour of tree search-based algorithms for POMDPs.
    - Unlike traditional methods, we augment belief tracking by approximating the uncertainty of transitioning into the current belief state given the real action and the real observation received from the world, denoted as $\tilde{P}(z|h_r, a)$ (Line:169).
    - Notably, this is a lightweight approach and adaptable to various tree search-based solutions, making it a versatile enhancement strategy.
- **Traditional methods**
    - Our particle resampling strategy is guided by the framework's execution, driven by $\tilde{P}(z|h_r, a)$, which hinges on I-UCB, being fundamentally different from the traditional resampling strategy in the literature. Our Ablation Study (Section 6) showcases the critical role of this connection in performance enhancement.
    - Distinguishing ourselves to other entropy-based methods, such as KLD, our particle filter update operates independently of extensive entropy or likelihood computations or the need for prior knowledge to define target distributions.

- **$\alpha$ function proposal:**
    - The $\alpha$ value, updated at the root level, influences action selection across all nodes.
    - Our update strategy integrates complete simulation information to adjust $\alpha$, aggregating all rewards and observations obtained throughout the simulation path. This update occurs after the entire observation (information) set is backpropagated to the root.
    - The backpropagation of observation is a novel strategy to maintain all observations (information) aggregated since the beginning of the search process, at the root node level.
    -  Addressing the challenge of estimating entropy in these algorithms, while handling unknown distributions and looking for the optimal actions, the novel $\alpha$ function adeptly manages the scenarios information to prevent problems (like preventing entropy overestimation due to expanding observation space (Line:248)), without relying on extensive computations or prior knowledge.

- **I-UCB design:**
    - I-UCB represents a novel variation of the traditional UCB function that smartly incorporates the concept of information gain into the tree search process.
    - Using the $\alpha$ function, I-UCB balances the trade-off between exploitation, exploration, and information value in an original and sophisticated manner
    - I-UCB can guide the search process and allow the algorithm to identify promising states even when rewards are not available or distributed sparsely within the planning horizon. This is novel since most of the UCB variants are focused on strategies that enhance the planning/search process based on rewards optimisation.

### Relevance/Significance
- **POMCP and UCB:** While our study centers on the enhancement of POMCP and the UCB equation, its applicability readily extends to diverse planning frameworks. It's noteworthy that both POMCP and UCB are still significant references in the literature [1,2,3] and consistently yielding impactful advancements in the field [4,5,6].
- **Weak constraints to run:** We refined action selection (via I-UCB) and proposed a novel framework (IB-POMCP) that *does not demand prior data collection or the modelling of latent functions*.
- **Handles sparse rewards:** IB-POMCP leverages entropy to estimate information gain using both real-world and internally generated observations, allowing it to identify promising states even *when rewards are not* readily *available* or *sparse* within the planning horizon.
- **Light-weight framework:** we *enhance planning without increasing reasoning time*. Our novel approach potentially impacts AI and decision-making by improving uncertainty planning capabilities while maintaining efficiency.

### Theoretical Analysis
- We provide the complete theoretical analysis of our proposal.
- In our Technical Appendix, we show that IB-POCMP convergence to epsilon-optimal action under a finite number of iterations.
- We analyse the convergence for the $\alpha$ function and I-UCB equation.
- All the theoretical guarantees are supported by our empirical results.

### Baselines and benchmarks
- We emphasise the our domains complexity comparing to traditional benchmark, presenting specific challenges and enabling different analysis.
    - The Maze problem allows us to study how IB-POMCP handles uncertainty and approximates belief to the actual world, as it involves self-localization challenges.
    - On the other hand, the Foraging environment enables us to explore how IB-POMCP deals with reward sparsity while planning under uncertainty.
- Our benchmark scenarios are far from toy problem complexity.
    - Our Foraging Warehouse scenario, for example, which has dimensions $20\times20$ and $|Z|$ in the order of $10^19$, showcases the scalability of IB-POMCP while surpassing the traditional LaserTag ($7\times11$ with $|Z|$ in the order of $10^6$) as presented in the referred DESPOT's paper.

[1] Sunberg, Z. et. al. "Online algorithms for POMDPs with continuous state, action, and observation spaces." In  ICAPS 2018

[2] Hoerger M. et. al. "An On-Line POMDP Solver for Continuous Observation Spaces". In ICRA 2021

[3] Amini, S. et. al. POMCP-based decentralized spatial task allocation algorithms for partially observable environments. In Appl Intell 2023

[4] Mern, J. et. al. Bayesian Optimized Monte Carlo Planning. In AAAI 2021

[5] Yourdshahi, E et al. "On-line estimators for ad-hoc task execution: learning types and parameters of teammates for effective teamwork." In AAMAS 2022

[6] Schwartz, J. et .al. "Bayes-Adaptive Monte-Carlo Planning for Type-Based Reasoning in Large Partially Observable, Multi-Agent Environments." In AAMAS 2023

---

### Decision · Program_Chairs · 2023-09-21

**Decision:**

Accept (poster)

**Comment:**

This paper presents a novel extension of POMCP for online planning in POMDPs. The approach (IB-POMCP) uses belief entropy to guide search, improving the handling of sparse rewards.

Online planning in POMDPs is a very difficult problem and while there has been work in the area, developing scalable and well-performing methods is still an open problem. This paper builds on the popular POMCP and UCB methods, making the approach widely applicable.

Where there was some concern about the experimental results and the generality of the method, the approach improves performance while maintaining strong guarantees. Furthermore, the author response with additional clarifications and experiments was very helpful. These responses were helpful for better understanding the novelty and significance of the method, understanding the theory, and confirming improved performance on other standard domains. The paper should include these updates in any final version.